# Exotic $\mathbb{Z}_N$ Symmetries, Duality, and Fractons in $3+1$-Dimensional Quantum Field Theory

Nathan Seiberg and Shu-Heng Shao

*School of Natural Sciences, Institute for Advanced Study,*
*Princeton, NJ 08540, USA*

**Abstract**

Following our earlier analyses of nonstandard continuum quantum field theories, we study here gapped systems in $3 + 1$ dimensions, which exhibit fractonic behavior. In particular, we present three dual field theory descriptions of the low-energy physics of the X-cube model. A key aspect of our constructions is the use of discontinuous fields in the continuum field theory. Spacetime is continuous, but the fields are not.

# 1   Introduction

The many diverse applications of quantum field theory make its general study interesting in its own right. Our investigation here was motivated by certain lattice systems. In this context continuum quantum field theory gives a universal description of the long-distance physics. As such, it is insensitive to most of the specific details of the microscopic model and therefore it captures its more generic aspects.

This paper is the third in a series of three papers (the earlier ones are [1] and [2]) exploring subtle continuum quantum field theories. (A followup paper [3] explores additional models.) This exploration was motivated by the recent exciting discovery of fractons (for reviews, see e.g. [4,5] and references therein), which exhibit phenomena that appear to be outside the scope of standard continuum quantum field theory.

Our continuum quantum field theories go beyond the standard framework in three ways:

- Not only are these quantum fields theories not Lorentz invariant, they are also not rotational invariant. The continuum limit is translation invariant, but it preserves only the finite rotation group of the underlying lattice. In [2] and in this paper only the $S_4$ subgroup of the $SO(3)$ rotations is preserved. This is the group generated by 90 degree rotations. The representations of this group are reviewed in Appendix A.

- As started in the analysis of such systems in [6] and continued in [1,2], the organizing principle of the discussion is the global symmetries of these systems. We refer to these symmetries, which are quite different than ordinary global symmetries, as exotic global symmetries. The discussion in [2] analyzed many such exotic symmetries and focused on four special ones. We review them in Appendix B and summarize them in Table 1.[1]

---

[1]As in [1,2], we limit ourselves to flat spacetime. Space will be mostly a rectangular three-torus $\mathbb{T}^3$. The signature will be either Lorentzian or Euclidean. And when it is Euclidean we will also consider the case of a rectangular four-torus $\mathbb{T}^4$. We will use $x^i$ with $i = 1, 2, 3$ or $x, y, z$ to denote the three spatial coordinates, $x^0$ or $t$ for Lorentzian time, and $\tau$ for Euclidean time. When space is a three-torus, the lengths of its three sides will be denoted by $\ell^x$, $\ell^y$, $\ell^z$. When we take an underlying lattice into account the number of sites in the three directions are $L^i = \frac{\ell^i}{a}$, where $a$ is the lattice spacing.

|  | $(\mathbf{2},\mathbf{3'})$ tensor symmetry | $(\mathbf{3'},\mathbf{2})$ tensor symmetry |
|---|---|---|
| symmetry operators | slab in the $xy$ plane between $z_1$ and $z_2$ $$\mathcal{U}^{xy}(z_1, z_2) \quad \text{etc.}$$ | line at $x, y$ along $z$ $$\mathcal{U}^z(x,y) \quad \text{etc.}$$ |
| constraints | $$\mathcal{U}^{xy}(0,\ell^z)\,\mathcal{U}^{yz}(0,\ell^x)\,\mathcal{U}^{zx}(0,\ell^y) = 1$$ | $\mathcal{U}^x(y,z) = \mathcal{U}^x_y(y)\,\mathcal{U}^x_z(z)$ $\mathcal{U}^y(z,x) = \mathcal{U}^y_z(z)\,\mathcal{U}^y_x(x)$ $\mathcal{U}^z(x,y) = \mathcal{U}^z_x(x)\,\mathcal{U}^z_y(y)$ |
| number of operators | $L^x + L^y + L^z - 1$ | $2L^x + 2L^y + 2L^z - 3$ |
|  | $(\mathbf{1},\mathbf{3'})$ dipole symmetry | $(\mathbf{3'},\mathbf{1})$ dipole symmetry |
| symmetry operators | slab in the $xy$ plane between $z_1$ and $z_2$ $$\mathcal{U}_{xy}(z_1, z_2) \quad \text{etc.}$$ | strip between $z_1$ and $z_2$ along a curve $\mathcal{C}^{xy}$ in the $xy$ plane $$\mathcal{U}(z_1, z_2, \mathcal{C}^{xy}) \quad \text{etc.}$$ |
| constraints | $$\mathcal{U}_{yz}(0,\ell^x) = \mathcal{U}_{zx}(0,\ell^y) = \mathcal{U}_{xy}(0,\ell^z)$$ | $\mathcal{U}(0,\ell^z,\mathcal{C}^{xy}_x) = \mathcal{U}(0,\ell^x,\mathcal{C}^{yz}_z)$ $\mathcal{U}(0,\ell^z,\mathcal{C}^{xy}_y) = \mathcal{U}(0,\ell^y,\mathcal{C}^{xz}_z)$ $\mathcal{U}(0,\ell^x,\mathcal{C}^{yz}_y) = \mathcal{U}(0,\ell^y,\mathcal{C}^{xz}_x)$ depends on $\mathcal{C}^{ij}$ topologically |
| number of operators | $L^x + L^y + L^z - 2$ | $2L^x + 2L^y + 2L^z - 3$ |

Table 1: The symmetry operators of the tensor and dipole global symmetries. We also show the number of these operators when we discretize the space to a lattice with $L^x, L^y, L^z$ sites in the three directions. For simplicity we wrote the symmetry operators for a single direction and added "etc." to represent the other directions. Every slab operator with argument $(0, \ell^x)$ etc. acts on the whole space. We further define the $(\mathbf{3'},\mathbf{2})$ unconstrained tensor and the $(\mathbf{3'},\mathbf{1})$ unconstrained dipole symmetries by relaxing the constraints above. See [2] and Appendix B for more details.

- The most significant departure from standard continuum quantum field theory is the use of discontinuous fields. The underlying spacetime is continuous, but we allow certain discontinuous field configurations. A crucial part of the analysis is the precise characterization of the allowed discontinuities. Since we discuss also gauge theories, we should pay attention to the allowed discontinuities in the gauge parameters, which determine the transition functions and the allowed twisted bundles.

All the systems in [1, 2] and here are natural in the sense that they include all low derivative terms that respect their specified global symmetries. However, an important point, which was stressed in [1,2], is that in these exotic systems, the notions of naturalness and universality are subtle.[2] Since we allow some discontinuous field configurations, the expansion in powers of derivatives might not be valid, and certain higher-derivative terms can be as significant as low-derivative terms. As a result, computations using the minimal Lagrangian might not lead to universal results.

A related issue is that of robustness. Most of the systems in [1, 2] are not robust. By that we mean that the low-energy theory includes relevant operators violating the global symmetry. Therefore, small changes in the short-distance physics deform the long-distance theory by these relevant operators. This ruins the elaborate long-distance physics of the system. (See [1] for a review of the role of naturalness and robustness in quantum field theory.)

However, it is important that all the models in this paper are both natural and robust. The low-energy theory does not have any local operators at all. This means that it does not have higher derivative operators that can ruin the universality and it does not have symmetry violating operators that can ruin its robustness. Therefore, small changes of the short-distance physics, including changes that break explicitly the global symmetry, cannot change the long-distance physics.

In [2] we studied four theories. Two of them are non-gauge theories, the $\phi$-theory and the $\hat{\phi}$-theory. The dynamical field $\phi$ is invariant under rotations, while $\hat{\phi}$ is in the two dimensional representation of the cubic group. Each of these theories has its own momentum and winding symmetries. The $\phi$-theory has a $U(1)$ $(\mathbf{1}, \mathbf{3}')$ dipole momentum symmetry and a $U(1)$ $(\mathbf{3}', \mathbf{1})$ dipole winding symmetry (see Table 2). The $\hat{\phi}$-theory has a $U(1)$ $(\mathbf{2}, \mathbf{3}')$ momentum tensor symmetry and a $U(1)$ $(\mathbf{3}', \mathbf{2})$ winding tensor symmetry (see Table 3).

Then we studied two gauge theories. The gauge symmetry of the $A$-theory is the momentum global symmetry of the $\phi$-theory, i.e. it is a $U(1)$ $(\mathbf{1}, \mathbf{3}')$ dipole symmetry. And the gauge symmetry of the $\hat{A}$-theory is the momentum global symmetry of the $\hat{\phi}$-theory, i.e. it is a $U(1)$ $(\mathbf{2}, \mathbf{3}')$ tensor symmetry. Some aspects of the $A$-theory had been discussed in [7–12]

---

[2]We thank P. Gorantla and H.T. Lam for useful discussions about these points.

| | | |
|---|---|---|
| Lagrangian | $\frac{\mu_0}{2}(\partial_0\phi)^2 - \frac{1}{4\mu}(\partial_i\partial_j\phi)^2$ | $\frac{1}{2\hat{g}_e^2}\hat{E}_{ij}\hat{E}^{ij} - \frac{1}{\hat{g}_m^2}\hat{B}^2$ |
| | | $\hat{E}^{ij} = \partial_0\hat{A}^{ij} - \partial_k\hat{A}_0^{k(ij)}$ $\hat{B} = \frac{1}{2}\partial_i\partial_j\hat{A}^{ij}$ |
| $(\mathbf{1}, \mathbf{3'})$ dipole symmetry | momentum $(J_0 = \mu_0\partial_0\phi, J^{ij} = -\frac{1}{\mu}\partial^i\partial^j\phi)$ | magnetic $(J_0 = \frac{1}{2\pi}\hat{B}, J^{ij} = \frac{1}{2\pi}\hat{E}^{ij})$ |
| currents | $\partial_0 J_0 = \frac{1}{2}\partial_i\partial_j J^{ij}$ | |
| charges | $Q_{xy}(z) = \oint dx \oint dy J_0 = \sum_\gamma W_{z\gamma}\delta(z - z_\gamma)$ $\oint dz Q_{xy}(z) = \oint dy Q_{zx}(y) = \oint dx Q_{yz}(x)$ | |
| energy | $\mathcal{O}(1/a)$ | |
| number of sectors | $L^x + L^y + L^z - 2$ | |
| $(\mathbf{3'}, \mathbf{1})$ dipole symmetry | winding $(J_0^{ij} = \frac{1}{2\pi}\partial^i\partial^j\phi, J = \frac{1}{2\pi}\partial_0\phi)$ | electric $(J_0^{ij} = -\frac{2}{\hat{g}_e^2}\hat{E}^{ij}, J = \frac{2}{\hat{g}_m^2}\hat{B})$ |
| currents | $\partial_0 J_0^{ij} = \partial^i\partial^j J$ $\partial^i J_0^{jk} = \partial^j J_0^{ik}$ | |
| charges | $Q(\mathcal{C}_i^{xy}, z) = \oint_{\mathcal{C}_i^{xy} \in (x,y)} (dx J_0^{zx} + dy J_0^{zy}) = \sum_\gamma W_{i\gamma}^z\delta(z - z_\gamma)$ $\oint dz Q(\mathcal{C}_x^{xy}, z) = \oint dx Q(\mathcal{C}_z^{yz}, x)$ | |
| energy | $\mathcal{O}(1/a)$ | |
| number of sectors | $2L^x + 2L^y + 2L^z - 3$ | |
| duality map | $\mu_0 = \frac{\hat{g}_m^2}{8\pi^2}$ $\qquad$ $\frac{1}{\mu} = \frac{\hat{g}_e^2}{8\pi^2}$ | |

Table 2: Global symmetries of the $U(1)$ tensor gauge theory $\hat{A}$ and its dual $\phi$. Here $\mathcal{C}_i^{ij}$ is a curve on the $ij$ plane that wraps around the $i$ cycle once but not the $j$ cycle. Above we have only shown charges for some directions, while the others admit similar expressions. See [2] for more details.

| Lagrangian | $\frac{\hat{\mu}_0}{12}(\partial_0\hat{\phi}^{i(jk)})^2 - \frac{\hat{\mu}}{2}(\partial_k\hat{\phi}^{k(ij)})^2$ | $\frac{1}{2g_e^2}E_{ij}E^{ij} - \frac{1}{2g_m^2}B_{[ij]k}B^{[ij]k}$ |
|---|---|---|
| | | $E_{ij} = \partial_0 A_{ij} - \partial_i\partial_j A_0$ |
| | | $B_{[ij]k} = \partial_i A_{jk} - \partial_j A_{ik}$ |
| $(\mathbf{2},\mathbf{3'})$ tensor symmetry | momentum $(J_0^{[ij]k} = \hat{\mu}_0\partial_0\hat{\phi}^{[ij]k}, J^{ij} = \hat{\mu}\partial_k\hat{\phi}^{k(ij)})$ | magnetic $(J_0^{[ij]k} = \frac{1}{2\pi}B^{[ij]k}, J^{ij} = \frac{1}{2\pi}E^{ij})$ |
| currents | $\partial_0 J_0^{[ij]k} = \partial^i J^{jk} - \partial^j J^{ik}$ | |
| charges | $Q^{[xy]}(z) = \oint dx \oint dy J_0^{[xy]z} = \sum_\gamma W_{z\gamma}\delta(z - z_\gamma)$ $\oint dz Q^{[xy]} + \oint dx Q^{[yz]x} + \oint dy Q^{[zx]y} = 0$ | |
| energy | $\mathcal{O}(1/a)$ | |
| number of sectors | $L^x + L^y + L^z - 1$ | |
| $(\mathbf{3'},\mathbf{2})$ tensor symmetry | winding $(J_0^{ij} = \frac{1}{2\pi}\partial_k\hat{\phi}^{k(ij)}, J^{k(ij)} = \frac{1}{2\pi}\partial_0\hat{\phi}^{k(ij)})$ | electric $(J_0^{ij} = \frac{2}{g_e^2}E^{ij}, J^{[ki]j} = \frac{2}{g_m^2}B^{[ki]j})$ |
| currents | $\partial_0 J_0^{ij} = \partial_k(J^{[ki]j} + J^{[kj]i})$ $\partial_i\partial_j J_0^{ij} = 0$ | |
| charges | $Q^z(x, y) = \oint dz J_0^{xy} = W_z^x(x) + W_z^y(y)$ $(W_z^x(x), W_z^y(y)) \sim (W_z^x(x) + 1, W_z^y(y) - 1)$ | |
| energy | $\mathcal{O}(a)$ | |
| number of sectors | $2L^x + 2L^y + 2L^z - 3$ | |
| duality map | $\hat{\mu}_0 = \frac{g_m^2}{8\pi^2}$ $\qquad$ $\hat{\mu} = \frac{g_e^2}{8\pi^2}$ | |

Table 3: Global symmetries of the $U(1)$ tensor gauge theory $A$ and its dual $\hat{\phi}$. Above we have only shown charges for some directions, while the others admit similar expressions. See [2] for more details.

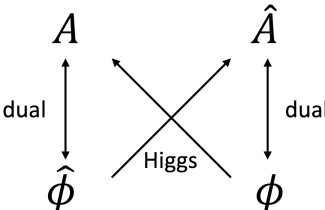

Figure 1: Relations between the four theories in [2]. The $U(1)$ tensor gauge theory of $A$ is dual to the non-gauge $\hat{\phi}$-theory, while the $U(1)$ tensor gauge theory of $\hat{A}$ is dual to the non-gauge $\phi$-theory. The $\phi$-theory is the Higgs field for the $U(1)$ gauge symmetry of $A$, while the $\hat{\phi}$-theory is the Higgs field for the $U(1)$ gauge symmetry of $\hat{A}$.

(see [13–28] for related tensor gauge theories). And some aspects of the $\hat{A}$-gauge theory had been discussed in [8]. In the absence of charged matter fields, these gauge theories have their own electric and magnetic global symmetries, which are similar to the electric and magnetic one-form global symmetries of ordinary $3+1$-dimensional gauge theories [29]. See Table 2 and Table 3.

Surprisingly, the $A$-theory turns out to be dual to the $\hat{\phi}$-theory and the $\hat{A}$-theory turns out to be dual to the $\phi$-theory. In every one of these dual pairs the global symmetries and the spectra match across the duality. See Table 2 and Table 3. This is particularly surprising given the subtle nature of the states that are charged under the momentum and winding symmetries of the non-gauge systems and states that are charged under the magnetic and electric symmetries of the gauge systems.

The relation between these four theories is summarized in Figure 1. See [2] for more details.

## *Outline*

In this paper we study the $\mathbb{Z}_N$ versions of these two gauge theories. We construct them by adding matter fields with charge $N$ to the $U(1)$ gauge theories and then Higgsing them to $\mathbb{Z}_N$. In Section 2 we use the fact that the gauge symmetry of the $A$-theory is the momentum symmetry of the $\phi$-theory to add charge-$N$ $\phi$ fields that Higgs it to $\mathbb{Z}_N$. Similarly, in Section 3 we add charge-$N$ $\hat{\phi}$ fields to Higgs the $\hat{A}$-theory to $\mathbb{Z}_N$. This is summarized in Figure 1.

Another convenient description of a $\mathbb{Z}_N$ continuum gauge theory is in terms of a $BF$-theory [30–32, 29]. We use this description in Section 4, which only involves the $A$ and the $\hat{A}$ gauge fields but not the $\phi$ or $\hat{\phi}$ fields. Certain aspects of this $BF$-type theory have been discussed in [8].

In Section 5, we show that the three different continuum theories in Sections 2, 3, and 4 are dual to each other. We will call these continuum field theories the $\mathbb{Z}_N$ tensor gauge theory.

The continuum $\mathbb{Z}_N$ tensor gauge theory describes the low-energy dynamics and the defects of the celebrated X-cube model [33] (see also [34]). In particular, it captures the restricted mobility of probe particles and the large ground state degeneracy of the X-cube model.

Crucial aspects of the analysis here rely on the understanding of the space of functions and the space of gauge fields in the four theories in [2]. This information determines which bundles are allowed, quantizes the coefficients in the Lagrangians, in the defects, and in the operators, and fixes the number of ground states.

As we said above, Appendix A reviews the representations of the cubic group and our notation and Appendix B reviews some aspects of the exotic global symmetries of [2].

Appendix C reviews the lattice description of $\mathbb{Z}_N$ gauge theories and their toric code presentation [35] and compares them with the continuum description of these theories. Even though this material is well known, we thought it would be helpful to present it here in order to clarify our perspective and to compare our various constructions to the known constructions of this well-studied case.

More specifically, our $\mathbb{Z}_N$ lattice gauge theories of $A$ and $\hat{A}$ are similar to ordinary lattice $\mathbb{Z}_N$ gauge theories, while the X-cube model is analogous to the toric code. The ordinary $\mathbb{Z}_N$ gauge theory and the toric code are dual in the low energy, which is described by the continuum $\mathbb{Z}_N$ gauge theory. Analogously, our lattice theories of $A$, $\hat{A}$ and the X-cube model are dual to each other at long distances, which is captured by the continuum $\mathbb{Z}_N$ tensor gauge theory.

# 2 $\mathbb{Z}_N$ Tensor Gauge Theory $A$

## 2.1 The Lattice Model

The first lattice tensor gauge theory is the $\mathbb{Z}_N$ version of the $U(1)$ lattice gauge theory of $A$ in [1]. The X-cube model [33] on the dual lattice can be viewed as a limit of this lattice gauge theory with Gauss law dynamically imposed.

We start with a Euclidean lattice and label the sites by integers $(\hat{\tau}, \hat{x}, \hat{y}, \hat{z})$. Let $L^i$ be the number of sites along the $x^i$ direction. As in standard lattice gauge theory, the gauge transformations are $\mathbb{Z}_N$ phases $\eta(\hat{\tau}, \hat{x}, \hat{y}, \hat{z})$ on the sites. The gauge fields are $\mathbb{Z}_N$ phases placed on the (Euclidean) temporal links $U_\tau$ and on the spatial plaquettes $U_{xy}$, $U_{xz}$, $U_{yz}$.

Note that there are no diagonal components of the gauge fields $U_{xx}, U_{yy}, U_{zz}$ associated with the sites. The authors of [10] referred to a theory without these variables as a "hollow gauge theory."

The gauge transformations act on them as

$$U_\tau(\hat{\tau}, \hat{x}, \hat{y}, \hat{z}) \to U_\tau(\hat{\tau}, \hat{x}, \hat{y}, \hat{z})\eta(\hat{\tau}, \hat{x}, \hat{y}, \hat{z})\eta(\tau + 1, \hat{x}, \hat{y}, \hat{z})^{-1},$$
$$U_{xy}(\hat{\tau}, \hat{x}, \hat{y}, \hat{z}) \to U_{xy}(\hat{\tau}, \hat{x}, \hat{y}, \hat{z})\eta(\hat{\tau}, \hat{x}, \hat{y}, \hat{z})\eta(\hat{\tau}, \hat{x} + 1, \hat{y}, \hat{z})^{-1}\eta(\hat{\tau}, \hat{x} + 1, \hat{y} + 1, \hat{z})\eta(\hat{\tau}, \hat{x}, \hat{y} + 1, \hat{z})^{-1},$$
$$(2.1)$$

and similarly for $U_{xz}$ and $U_{yz}$. The Euclidean time-like links have standard gauge transformation rules (see (C.1)) and the plaquette elements are multiplied by the 4 phases around the plaquette.

The lattice action can include many gauge invariant terms. The simplest ones are associated with cubes in the time-space-space directions and in the space-space-space directions

$$L_{xy\tau}(\hat{\tau}, \hat{x}, \hat{y}, \hat{z}) = U_\tau(\hat{\tau}, \hat{x}, \hat{y}, \hat{z})U_\tau(\hat{\tau}, \hat{x} + 1, \hat{y}, \hat{z})^{-1}U_\tau(\hat{\tau}, \hat{x} + 1, \hat{y} + 1, \hat{z})U_\tau(\hat{\tau}, \hat{x}, \hat{y} + 1, \hat{z})^{-1}$$
$$U_{xy}(\hat{\tau}, \hat{x}, \hat{y}, \hat{z})^{-1}U_{xy}(\hat{\tau} + 1, \hat{x}, \hat{y}, \hat{z})$$
$$L_{[zx]y}(\hat{\tau}, \hat{x}, \hat{y}, \hat{z}) = U_{xy}(\hat{\tau}, \hat{x}, \hat{y}, \hat{z} + 1)U_{xy}(\hat{\tau}, \hat{x}, \hat{y}, \hat{z})^{-1}U_{yz}(\hat{\tau}, \hat{x} + 1, \hat{y}, \hat{z})^{-1}U_{yz}(\hat{\tau}, \hat{x}, \hat{y}, \hat{z})$$
$$(2.2)$$

and similarly for the other directions.

In addition to the local gauge-invariant operators (2.2), there are other non-local, extended ones. One example is a "strip" on the $xz$-plane:

$$W(\mathcal{C}^{xy}) = \prod_{\hat{x}=1}^{L^x} U_{xz}(\hat{\tau}, \hat{x}, \hat{y}, \hat{z}), \qquad (2.3)$$

and similarly for the other components in other directions. Here $\mathcal{C}^{xy}$ denotes the constant $\hat{y}$ line on the $xy$-plane. More generally, the strip can be made out of plaquettes extending between $\hat{z}$ and $\hat{z} + 1$ and zigzagging along a path $\mathcal{C}^{xy}$ on the $xy$-plane.

In the Hamiltonian formulation, we choose the temporal gauge to set all the $U_\tau$'s to 1. Let $V_{ij}$ be the conjugate momenta $V_{ij}$ of $U_{ij}$. They obey the $\mathbb{Z}_N$ Heisenberg algebra[3] $U_{ij}V_{ij} = e^{2\pi i/N}V_{ij}U_{ij}$ if they belong to the same plaquette, and commute otherwise.

Gauss law is imposed as an operator equation

$$G(\hat{x}, \hat{y}, \hat{z}) = \prod_{p \ni (\hat{x}, \hat{y}, \hat{z})} V_p^{\epsilon_p} = 1 \qquad (2.4)$$

---

[3]By $\mathbb{Z}_N$ Heisenberg algebra, we mean the algebra generated by the clock and shift operators $A$, $B$ satisfying $A^N = B^N = 1$ and $AB = e^{2\pi i/N}BA$. This algebra arises in many different contexts and has many names.

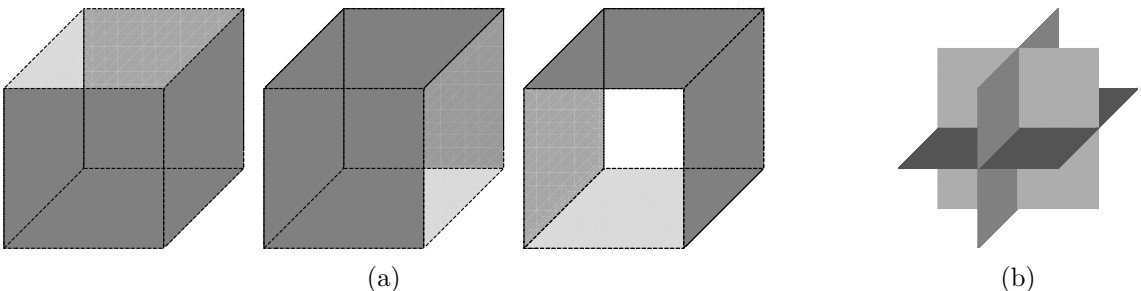

Figure 2: (a) The term $L_{[xy]z}, L_{[zx]y}, L_{[yz]x}$ in the Hamiltonian, which are products of the $U$'s on the plaquettes. (b) Gauss law constraint $G$, which is a products of 12 $V$'s on the plaquettes. The shaded faces stand for $U$ and $V$ on the plaquette in (a) and (b), respectively. We suppress the orientation of these plaquette variables.

where the product is an oriented product ($\epsilon_p = \pm 1$) over the 12 plaquettes $p$ that share a common site $(\hat{x}, \hat{y}, \hat{z})$.

The Hamiltonian is

$$H = -\frac{1}{g_e^2} \sum_{\text{plaquettes}} V - \frac{1}{g_m^2} \sum_{\text{cubes}} \left( L_{[xy]z} + L_{[yz]x} + L_{[zx]y} \right) + c.c.. \tag{2.5}$$

The lattice model has a $\mathbb{Z}_N$ electric tensor symmetry whose conserved symmetry operator[4] is proportional to

$$\prod_{\hat{z}=1}^{L^z} V_{xy}(\hat{x}_0, \hat{y}_0, \hat{z}), \tag{2.6}$$

for each point $(\hat{x}_0, \hat{y}_0)$ on the $xy$-plane. There are similar symmetry operators along the other directions. This symmetry operator commutes with the Hamiltonian, in particular the $L_{[ij]k}$ terms. The electric tensor symmetry rotates the plaquette variables $U_{xy}$ at $(\hat{x}_0, \hat{y}_0)$ for all $\hat{z}$ by a $\mathbb{Z}_N$ phase. Using Gauss law (2.4), the dependence of the conserved operator on $p$ is a function of $\hat{x}_0$ times a function of $\hat{y}_0$.

As in the standard $\mathbb{Z}_N$ gauge theory, instead of imposing the Gauss law as an operator equation, we can alternatively impose it energetically by adding a term $-\sum_{\text{sites}} G$ to the Hamiltonian. One example of such Hamiltonian is

$$H = -\frac{1}{g_e^2} \sum_{\text{plaquettes}} V - \frac{1}{g^2} \sum_{\text{sites}} G - \frac{1}{g_m^2} \sum_{\text{cubes}} \left( L_{[xy]z} + L_{[yz]x} + L_{[zx]y} \right) + c.c. \tag{2.7}$$

---

[4]When we discussed continuous symmetries, we used the phrase "charge" for the generator of infinitesimal transformations. Here, where the symmetry is discrete, we use "symmetry operator" for the generator of the symmetry.

The limit $g_e \to \infty$ gives the Hamiltonian of the X-cube model [33].

## 2.2   Continuum Lagrangian

We now present the continuum description for the $\mathbb{Z}_N$ lattice gauge theory of $A$ in Section 2.1. It is obtained by coupling the $U(1)$ tensor gauge theory of $A$ to a charge-$N$ Higgs field $\phi$. (See [2] for discussions on the $A$- and $\phi$-theories.) The Euclidean Lagrangian is:

$$\mathcal{L}_E = -\frac{i}{2(2\pi)}\hat{E}^{ij}(\partial_i\partial_j\phi - NA_{ij}) - \frac{i}{2\pi}\hat{B}(\partial_0\phi - NA_0)\,. \tag{2.8}$$

The fields $\hat{E}^{ij}$ in the $\mathbf{3'}$ and $\hat{B}$ in the $\mathbf{1}$ are Lagrangian multipliers. The $U(1)$ gauge transformation acts as

$$
\begin{aligned}
A_0 &\sim A_0 + \partial_0\alpha\,, &\quad A_{ij} &\sim A_{ij} + \partial_i\partial_j\alpha\,, \\
\phi &\sim \phi + N\alpha\,,
\end{aligned}
\tag{2.9}
$$

with $\alpha$ a $2\pi$-periodic gauge parameter.

The equations of motion are

$$
\begin{aligned}
\partial_i\partial_j\phi - NA_{ij} &= 0\,, \\
\partial_0\phi - NA_0 &= 0\,, \\
\hat{E}^{ij} = \hat{B} &= 0\,.
\end{aligned}
\tag{2.10}
$$

In particular, the equations of motion imply that the gauge-invariant field strengths of $A$ vanish:

$$
\begin{aligned}
E_{ij} &= \partial_0 A_{ij} - \partial_i\partial_j A_0 = 0\,, \\
B_{[ij]k} &= \partial_i A_{jk} - \partial_j A_{ik} = 0\,.
\end{aligned}
\tag{2.11}
$$

Since there is no local operator in this theory, the low energy field theory is robust (see [1] for a discussion of robustness). Similarly, since there are no local operators, the possible universality violation due to higher-derivative terms, which was discussed in [1, 2], is not present.

## 2.3   Global Symmetries

Let us track the global symmetries of the system from the $\phi$ and the $U(1)$ tensor gauge theory of $A$ in [2].

The scalar field theory of $\phi$ has a global $U(1)$ $(\mathbf{1}, \mathbf{3'})$ momentum dipole symmetry and a global $U(1)$ $(\mathbf{3'}, \mathbf{1})$ winding dipole symmetry. The momentum symmetry is gauged and

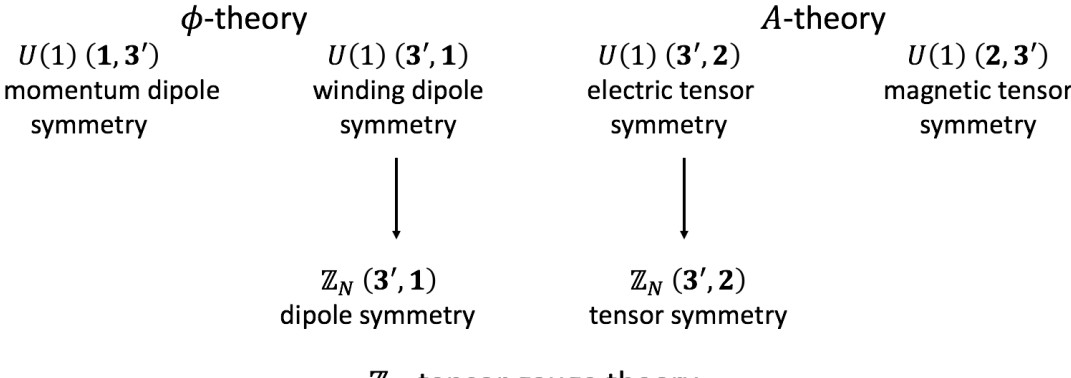

Figure 3: The global symmetries of the $U(1)$ $A$-theory, the $\phi$-theory, and the $\mathbb{Z}_N$ tensor gauge theory and their relations. The momentum dipole symmetry of the $\phi$-theory is gauged and therefore it is absent in the $\mathbb{Z}_N$ tensor gauge theory. The magnetic tensor symmetry of the $A$-theory is absent in the $\mathbb{Z}_N$ tensor gauge theory because of the constraint (2.11).

the gauging turns the $U(1)$ winding dipole symmetry into $\mathbb{Z}_N$. In addition, the pure gauge theory of $A$ has a $U(1)$ $(\mathbf{3'}, \mathbf{2})$ electric tensor symmetry and the coupling to the matter field $\phi$ breaks it to $\mathbb{Z}_N$. Altogether, we have a $\mathbb{Z}_N$ dipole global symmetry and a $\mathbb{Z}_N$ tensor global symmetry. See Figure 3.

The $\mathbb{Z}_N$ tensor symmetry is the electric symmetry on the lattice (2.6). Its symmetry operator cannot be written in terms of the fields in the Lagrangian (2.8) in a local way.

The $\mathbb{Z}_N$ dipole symmetry is not present on the lattice (2.7). In the continuum, its symmetry operator is a strip in space

$$
\begin{aligned}
W(z_1, z_2, \mathcal{C}^{xy}) &= \exp\left[i \int_{z_1}^{z_2} dz \oint_{\mathcal{C}^{xy}} (dx\, A_{xz} + dy\, A_{yz})\right] \\
&= \exp\left[\frac{i}{N} \int_{z_1}^{z_2} dz \oint_{\mathcal{C}^{xy}} (dx\, \partial_x \partial_z \phi + dy\, \partial_y \partial_z \phi)\right]
\end{aligned}
\tag{2.12}
$$

where we have used the equation of motion (2.10). Here $\mathcal{C}^{xy}$ is a closed curve on the $xy$-plane.

Only integer powers of the strip operator are invariant under the large gauge transformation

$$
\alpha = 2\pi \left[\frac{z}{\ell^z} \Theta(x - x_0) + \frac{x}{\ell^x} \Theta(z - z_0) - \frac{zx}{\ell^z \ell^x}\right] .
\tag{2.13}
$$

Furthermore, since the integral

$$\int_{z_1}^{z_2} dz \oint dx \partial_z \partial_x \phi \in 2\pi \mathbb{Z} \tag{2.14}$$

is the quantized winding dipole charge of the $\phi$-theory [2], we have $[W(z_1, z_2, \mathcal{C}^{xy})]^N = 1$. Therefore $W(z_1, z_2, \mathcal{C}^{xy})$ is a $\mathbb{Z}_N$ operator.

Similarly, there are strip operators along the other directions. They obey

$$
\begin{aligned}
W(0, \ell^z, \mathcal{C}_x^{xy}) &= W(0, \ell^x, \mathcal{C}_z^{yz}), \\
W(0, \ell^z, \mathcal{C}_y^{xy}) &= W(0, \ell^y, \mathcal{C}_z^{xz}), \\
W(0, \ell^x, \mathcal{C}_y^{yz}) &= W(0, \ell^y, \mathcal{C}_x^{xz}),
\end{aligned} \tag{2.15}
$$

where $\mathcal{C}_i^{ij}$ is a closed curve on the $ij$-plane that wraps around the $i$ direction once but not the $j$ direction. See Appendix B.2 for a more abstract discussion of the $\mathbb{Z}_N$ $(\mathbf{3'}, \mathbf{1})$ dipole symmetry.

In Section 4.2, we will discuss these symmetry operators and their associated defects in more details.

## 2.4   Ground State Degeneracy

From the equation of motion (2.10), we can solve all the other fields in terms of $\phi$, and the solution space reduces to

$$\left\{ \phi \right\} / \phi \sim \phi + N\alpha. \tag{2.16}$$

In particular, there is no local excitation in the $\mathbb{Z}_N$ tensor gauge theory.

Let us enumerate the number of states in this system. Almost all configurations of $\phi$ can be gauged away completely, except for the winding modes (see [1, 2] for details on these winding modes in the $\phi$-theory):

$$
\begin{aligned}
\phi(t, x, y) = {}& 2\pi \left[ \frac{x}{\ell^x} \sum_\beta W_{x\beta}^y \, \Theta(y - y_\beta) + \frac{y}{\ell^y} \sum_\alpha W_{y\alpha}^x \, \Theta(x - x_\alpha) - W^{xy} \frac{xy}{\ell^x \ell^y} \right] \\
&+ 2\pi \left[ \frac{x}{\ell^x} \sum_\gamma W_{x\gamma}^z \, \Theta(z - z_\gamma) + \frac{z}{\ell^z} \sum_\alpha W_{z\alpha}^x \, \Theta(x - x_\alpha) - W^{zx} \frac{zx}{\ell^z \ell^x} \right] \\
&+ 2\pi \left[ \frac{z}{\ell^z} \sum_\beta W_{z\beta}^y \, \Theta(y - y_\beta) + \frac{y}{\ell^y} \sum_\gamma W_{y\gamma}^z \, \Theta(z - z_\gamma) - W^{yz} \frac{yz}{\ell^y \ell^z} \right]
\end{aligned} \tag{2.17}
$$

where $W^i_{j\alpha} \in \mathbb{Z}$ and $W^{ij} = \sum_\alpha W^j_{i\alpha} = \sum_\beta W^i_{j\beta}$. On a lattice, these winding modes are labeled by $2L^x + 2L^y + 2L^z - 3$ integers. Similarly, the gauge parameter $\alpha$ can also have the above winding modes. Therefore, there are $N^{2L^x+2L^y+2L^z-3}$ winding modes that cannot be gauged away with their $W$ valued in $\mathbb{Z}_N$.

# 3  $\mathbb{Z}_N$ Tensor Gauge Theory $\hat{A}$

## 3.1  The Lattice Model

The X-cube model [33], with variables living on the links, can be viewed as a limit of another lattice gauge theory with Gauss law dynamically imposed. We now discuss this lattice gauge theory. Certain aspects of this lattice model have been discussed in [12].

We start with the Lagrangian formulation of this lattice model on a Euclidean lattice. The gauge parameters are $\mathbb{Z}_N$ phases placed on the sites. For each site $(\hat{\tau}, \hat{x}, \hat{y}, \hat{z})$, there are three gauge parameters $\hat{\eta}^{i(jk)}(\hat{\tau}, \hat{x}, \hat{y}, \hat{z})$ in the **2** satisfying $\hat{\eta}^{x(yz)}\hat{\eta}^{y(zx)}\hat{\eta}^{z(xy)} = 1$ at every site. (Recall that $i \neq j \neq k$.) The gauge fields are $\mathbb{Z}_N$ phases placed on the links. Associated with each temporal link, there are three gauge fields $\hat{U}_\tau^{i(jk)}(\hat{\tau}, \hat{x}, \hat{y}, \hat{z})$ in the **2** satisfying $\hat{U}_\tau^{x(yz)}\hat{U}_\tau^{y(zx)}\hat{U}_\tau^{z(xy)} = 1$. Associated with each spatial link along the $k$ direction, there is a gauge field $\hat{U}^{ij}$ in the **3′**.

The gauge transformations act on them as

$$
\begin{aligned}
\hat{U}_\tau^{i(jk)}(\hat{\tau}, \hat{x}, \hat{y}, \hat{z}) &\to \hat{U}_\tau^{i(jk)}(\hat{\tau}, \hat{x}, \hat{y}, \hat{z})\, \hat{\eta}^{i(jk)}(\hat{\tau}, \hat{x}, \hat{y}, \hat{z})\, \hat{\eta}^{i(jk)}(\hat{\tau}+1, \hat{x}, \hat{y}, \hat{z})^{-1}\,, \\
\hat{U}^{xy}(\hat{\tau}, \hat{x}, \hat{y}, \hat{z}) &\to \hat{U}^{xy}(\hat{\tau}, \hat{x}, \hat{y}, \hat{z})\, \hat{\eta}^{z(xy)}(\hat{\tau}, \hat{x}, \hat{y}, \hat{z})\, \hat{\eta}^{z(xy)}(\hat{\tau}, \hat{x}, \hat{y}, \hat{z}+1)^{-1}\,,
\end{aligned}
\tag{3.1}
$$

and similarly for $\hat{U}^{yz}$ and $\hat{U}^{zx}$.

Let us discuss the gauge invariant local terms in the action. The first kind is a plaquette on the $\tau z$-plane:

$$
\hat{L}^{\tau z}(\hat{\tau}, \hat{x}, \hat{y}, \hat{z}) = \hat{U}^{xy}(\hat{\tau}, \hat{x}, \hat{y}, \hat{z})\, \hat{U}_\tau^{z(xy)}(\hat{\tau}, \hat{x}, \hat{y}, \hat{z}+1)\, \hat{U}^{xy}(\hat{\tau}+1, \hat{x}, \hat{y}, \hat{z})^{-1}\, \hat{U}_\tau^{z(xy)}(\hat{\tau}, \hat{x}, \hat{y}, \hat{z})^{-1}
\tag{3.2}
$$

and similarly for $\hat{L}^{\tau x}$ and $\hat{L}^{\tau y}$. The second kind is a product of 12 spatial links around a

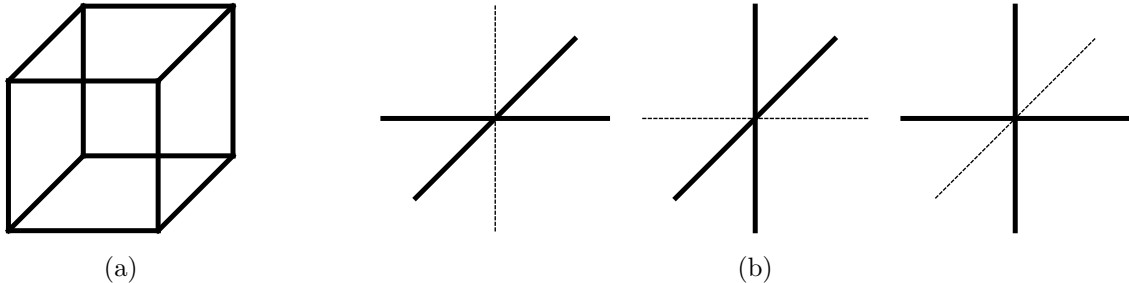

(a)                                                     (b)

Figure 4: (a) The term $\hat{L}$ in the Hamiltonian, which is a product of the $\hat{U}$'s of the 12 links around a cube. (b) The three Gauss law constraints $\hat{G}^{[xy]z} = \hat{G}^{[zx]y} = \hat{G}^{[yz]x} = 1$, which are products of the $\hat{V}$'s on the links. The solid lines stand for $\hat{U}$ and $\hat{V}$ on the link in (a) and (b), respectively. We suppress the orientation of these link variables.

cube in space at a fixed time:

$$
\begin{aligned}
\hat{L}(\hat{\tau}, \hat{x}, \hat{y}, \hat{z}) = {} & \hat{U}^{yz}(\hat{\tau}, \hat{x}, \hat{y}, \hat{z})\, \hat{U}^{zx}(\hat{\tau}, \hat{x}+1, \hat{y}, \hat{z})^{-1}\, \hat{U}^{yz}(\hat{\tau}, \hat{x}, \hat{y}+1, \hat{z})^{-1}\, \hat{U}^{zx}(\hat{\tau}, \hat{x}, \hat{y}, \hat{z}) \\
& \times \hat{U}^{yz}(\hat{\tau}, \hat{x}, \hat{y}, \hat{z}+1)^{-1}\, \hat{U}^{zx}(\hat{\tau}, \hat{x}+1, \hat{y}, \hat{z}+1)\, \hat{U}^{yz}(\hat{\tau}, \hat{x}, \hat{y}+1, \hat{z}+1)\, \hat{U}^{zx}(\hat{\tau}, \hat{x}, \hat{y}, \hat{z}+1)^{-1} \\
& \times \hat{U}^{xy}(\hat{\tau}, \hat{x}, \hat{y}, \hat{z})\, \hat{U}^{xy}(\hat{\tau}, \hat{x}+1, \hat{y}, \hat{z})^{-1}\, \hat{U}^{xy}(\hat{\tau}, \hat{x}+1, \hat{y}+1, \hat{z})\, \hat{U}^{xy}(\hat{\tau}, \hat{x}, \hat{y}+1, \hat{z})^{-1} \, .
\end{aligned}
\tag{3.3}
$$

The Lagrangian for this lattice model is a sum over the above terms.

In addition to the local, gauge-invariant operators (3.3), there are other non-local, extended ones. For example, we have a line operator along the $x^k$ direction.

$$
\prod_{\hat{x}^k=1}^{L^k} \hat{U}^{ij} \, .
\tag{3.4}
$$

In the Hamiltonian formulation, we choose the temporal gauge to set all the $\hat{U}_\tau^{i(jk)}$'s to 1. Let $\hat{V}^{ij}$ be the conjugate momenta for $\hat{U}^{ij}$. They obey the $\mathbb{Z}_N$ Heisenberg algebra $\hat{U}^{ij}\hat{V}^{ij} = e^{2\pi i/N}\hat{V}^{ij}\hat{U}^{ij}$ if they belong to the same link and commute otherwise. Gauss law is imposed as an operator equation

$$
\hat{G}^{[zx]y}(\hat{x}, \hat{y}, \hat{z}) = \hat{V}^{xy}(\hat{x}, \hat{y}, \hat{z}+1)\hat{V}^{xy}(\hat{x}, \hat{y}, \hat{z})^{-1}\hat{V}^{yz}(\hat{x}+1, \hat{y}, \hat{z})^{-1}\hat{V}^{yz}(\hat{x}, \hat{y}, \hat{z}) = 1
\tag{3.5}
$$

and similarly $\hat{G}^{[xy]z} = 1$ and $\hat{G}^{[yz]x} = 1$. Note that $\hat{G}^{[xy]z}\hat{G}^{[yz]x}\hat{G}^{[zx]y} = 1$ identically, so the Gauss law operator is in the **2**.

The Hamiltonian is

$$H = -\frac{1}{\hat{g}_e^2} \sum_{\text{links}} \hat{V} - \frac{1}{\hat{g}_m^2} \sum_{\text{cubes}} \hat{L} + c.c. \,. \tag{3.6}$$

The lattice model has an electric dipole symmetry whose charge operators are

$$\prod_{\hat{x}=1}^{L^x} \hat{V}^{zx}(\hat{x}, \hat{y}_0, \hat{z}_0), \quad \prod_{\hat{y}=1}^{L^y} \hat{V}^{zy}(\hat{x}_0, \hat{y}, \hat{z}_0). \tag{3.7}$$

There are 4 other operators associated with the other directions. They commute with the Hamiltonian, in particular the $\hat{L}$ terms. These two electric dipole symmetries rotate the phases of $\hat{U}^{ij}$ along a strip on the $zx$ and $yz$ planes, respectively.

Alternatively, we can impose Gauss law energetically by adding the following term to the Hamiltonian:

$$H = -\frac{1}{\hat{g}_e^2} \sum_{\text{links}} \hat{V} - \frac{1}{\hat{g}_m^2} \sum_{\text{cubes}} \hat{L} - \frac{1}{\hat{g}} \sum_{\text{sites}} (\hat{G}^{x(yz)} + \hat{G}^{y(zx)} + \hat{G}^{z(xy)}) + c.c. \,. \tag{3.8}$$

The limit $\hat{g}_e \to \infty$ gives the Hamiltonian of the X-cube model [33].

## 3.2   Continuum Lagrangian

We now present the continuum Lagrangian for the $\mathbb{Z}_N$ lattice gauge theory of $\hat{A}$ in Section 3.1. The Euclidean Lagrangian is:[5]

$$\mathcal{L}_E = \frac{i}{2(2\pi)} E_{ij} \left( \partial_k \hat{\phi}^{k(ij)} - N\hat{A}^{ij} \right) - \frac{i}{6(2\pi)} B_{k(ij)} \left( \partial_0 \hat{\phi}^{k(ij)} - N\hat{A}_0^{k(ij)} \right) \tag{3.10}$$

where $(\hat{A}_0^{k(ij)}, \hat{A}^{ij})$ are gauge fields in the $(\mathbf{2}, \mathbf{3'})$ of $S_4$ and $\hat{\phi}^{k(ij)}$ is a Higgs field in the $\mathbf{2}$ with charge $N$. $E_{ij}$ and $B_{[ij]k}$ are Lagrangian multipliers in the $\mathbf{3'}$ and $\mathbf{2}$ of $S_4$, respectively.

The gauge symmetry is

$$\hat{A}^{k(ij)} \sim \hat{A}^{k(ij)} + \partial_0 \hat{\alpha}^{k(ij)}, \qquad \hat{A}^{ij} \sim \hat{A}^{ij} + \partial_k \hat{\alpha}^{k(ij)},$$
$$\hat{\phi}^{k(ij)} \sim \hat{\phi}^{k(ij)} + N\hat{\alpha}^{k(ij)}. \tag{3.11}$$

---

[5]Recall that there are two presentations for a field in the $\mathbf{2}$ of $S_4$, $\hat{\phi}^{k(ij)}$ and $\hat{\phi}^{[ij]k}$ (see Appendix A). In the $\phi^{[ij]k}$ basis, the Lagrangian becomes

$$\mathcal{L}_E = \frac{i}{2(2\pi)} E_{ij} \left[ \partial_k \left( \hat{\phi}^{[ki]j} + \hat{\phi}^{[kj]i} \right) - N\hat{A}^{ij} \right] - \frac{i}{2(2\pi)} B_{[ij]k} \left( \partial_0 \hat{\phi}^{[ij]k} - N\hat{A}_0^{[ij]k} \right). \tag{3.9}$$

The equations of motion are

$$\partial_k \hat{\phi}^{k(ij)} - N\hat{A}^{ij} = 0\,,$$
$$\partial_0 \hat{\phi}^{k(ij)} - N\hat{A}_0^{k(ij)} = 0\,, \tag{3.12}$$
$$E^{ij} = B_{k(ij)} = 0\,.$$

In particular, the equations of motion imply that the gauge-invariant field strengths of $\hat{A}$ vanish:

$$\hat{E}^{ij} = \partial_0 \hat{A}^{ij} - \partial_k \hat{A}^{k(ij)} = 0\,,$$
$$\hat{B} = \frac{1}{2}\partial_i\partial_j \hat{A}^{ij} = 0\,. \tag{3.13}$$

In the above we have used $\partial_i\partial_j\partial_k \hat{\phi}^{k(ij)} = 0$.

Since there is no local operator in this theory, the low energy field theory is robust (see [1] for a discussion of robustness). Similarly, as above, since there are no local operators, the possible universality violation due to higher-derivative terms, which was discussed in [1, 2], is not present.

## 3.3  Global Symmetries

Let us track the global symmetries of the system from the $\hat{\phi}$ and the $U(1)$ tensor gauge theory of $\hat{A}$ in [2].

The field theory of $\hat{\phi}$ has a global $U(1)$ $(\mathbf{2}, \mathbf{3}')$ momentum tensor symmetry and a global $U(1)$ $(\mathbf{3}', \mathbf{2})$ winding tensor symmetry. The momentum symmetry is gauged and the gauging turns the $U(1)$ winding tensor symmetry into $\mathbb{Z}_N$. In addition, the pure gauge theory of $\hat{A}$ has a $U(1)$ $(\mathbf{3}', \mathbf{1})$ electric dipole symmetry and the coupling to the matter field $\hat{\phi}$ breaks it to $\mathbb{Z}_N$. Altogether, we have a $\mathbb{Z}_N$ dipole global symmetry and a $\mathbb{Z}_N$ tensor global symmetry. See Figure 5. This is the same global symmetry in Section 2.3 for the $\mathbb{Z}_N$ tensor gauge theory of $A$. Indeed, we will show that the two continuum $\mathbb{Z}_N$ tensor gauge theories of $\hat{A}$ and $A$ are dual to each other in Section 5.1.

The $\mathbb{Z}_N$ dipole symmetry is the electric symmetry on the lattice (3.7). Its symmetry operator cannot be written in terms of the fields in the Lagrangian (3.10) in a local way.

The $\mathbb{Z}_N$ tensor symmetry is not present on the lattice (3.8). In the continuum, its symmetry operator is a line in space

$$\hat{W}^z(x,y) = \exp\left[i\oint dz\hat{A}^{xy}\right] = \exp\left[\frac{i}{N}\oint dz\partial_z\hat{\phi}^{z(xy)}\right]\,, \tag{3.14}$$

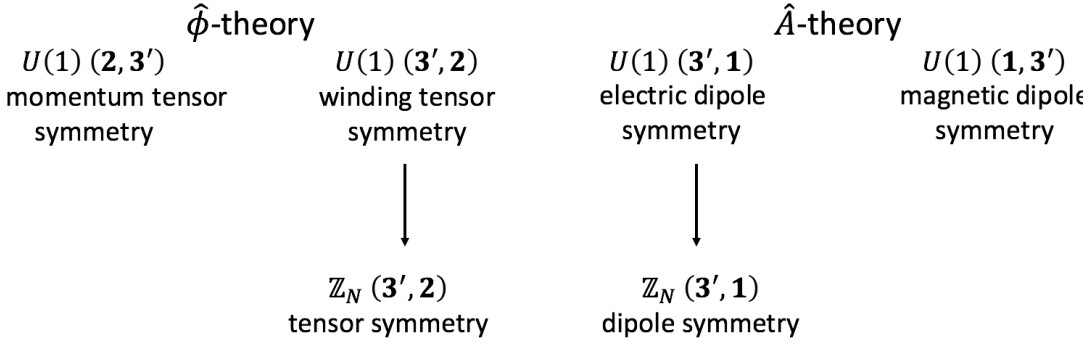

Figure 5: The global symmetries of the $U(1)$ $\hat{A}$-theory, the $\hat{\phi}$-theory, and the $\mathbb{Z}_N$ tensor gauge theory and their relations. The momentum tensor symmetry of the $\hat{\phi}$-theory is gauged and therefore it is absent in the $\mathbb{Z}_N$ tensor gauge theory. The magnetic dipole symmetry of the $\hat{A}$-theory is absent in the $\mathbb{Z}_N$ tensor gauge theory because of the constraint (3.13).

where we have used the equation of motion (3.12).

Only integer powers of $\hat{W}^z(x, y)$ are invariant under the large gauge transformation

$$\hat{\alpha}^{z(xy)} = -\hat{\alpha}^{y(zx)} = 2\pi \frac{z}{\ell^z}, \quad \hat{\alpha}^{x(yz)} = 0. \tag{3.15}$$

Furthermore, since the integral

$$\oint dz \partial_z \hat{\phi}^{z(xy)} \in 2\pi\mathbb{Z} \tag{3.16}$$

is the quantized winding tensor charge of the $\hat{\phi}$-theory [1], we have $[\hat{W}^z(x, y)]^N = 1$. Therefore $\hat{W}^z(x, y)$ is a $\mathbb{Z}_N$ operator.

Let us comment on the spatial dependence of $\hat{W}^z(x, y)$. Since $\hat{B} = 0$, we have

$$\partial_x \partial_y \oint dz \hat{A}^{xy} = -\oint dz \left( \partial_z \partial_x \hat{A}^{zx} + \partial_z \partial_y \hat{A}^{yz} \right) = 0. \tag{3.17}$$

Hence the dependence of the $\mathbb{Z}_N$ symmetry operator $\hat{W}^z(x, y)$ on $x, y$ factorizes

$$\hat{W}^z(x, y) = \hat{W}_x^z(x)\hat{W}_y^z(y). \tag{3.18}$$

Similarly, there are line operators along the other directions. See Appendix B.1 for a more abstract discussion of the $\mathbb{Z}_N$ $(\mathbf{3}', \mathbf{2})$ tensor symmetry.

In Section 4.2, we will discuss these symmetry operators and their associated defects in more details.

## 3.4 Ground State Degeneracy

From (3.12), we can solve all the other fields in terms of $\hat{\phi}^{k(ij)}$, and the solution space reduces to

$$\left\{ \hat{\phi}^{k(ij)} \right\} \; / \; \hat{\phi}^{k(ij)} \sim \hat{\phi}^{k(ij)} + N\hat{\alpha}^{k(ij)} \,. \tag{3.19}$$

Almost all configurations of $\hat{\phi}^{k(ij)}$ can be gauged away, except for the winding modes (see [2] for details on these winding modes):

$$\hat{\phi}^{x(yz)} = 2\pi \frac{x}{\ell^x} \left( W_x^y(y) + W_x^z(z) \right) - 2\pi \frac{W_y^z(z)y}{\ell^y} - 2\pi \frac{W_z^y(y)z}{\ell^z} \,,$$

$$\hat{\phi}^{y(zx)} = 2\pi \frac{y}{\ell^y} \left( W_y^z(z) + W_y^x(x) \right) - 2\pi \frac{W_z^x(x)z}{\ell^z} - 2\pi \frac{W_x^z(z)x}{\ell^x} \,, \tag{3.20}$$

$$\hat{\phi}^{z(xy)} = 2\pi \frac{z}{\ell^z} \left( W_z^x(x) + W_z^y(y) \right) - 2\pi \frac{W_y^x(x)y}{\ell^y} - 2\pi \frac{W_x^y(y)x}{\ell^x} \,,$$

where $W_j^i(x^i) \in \mathbb{Z}$. There is an identification

$$\begin{aligned} W_z^x(x) &\sim W_z^x(x) + 1 \,, \\ W_z^y(y) &\sim W_z^y(y) - 1 \,, \end{aligned} \tag{3.21}$$

and similarly for the other $W$'s. If we regularize the space by a lattice, these winding modes are labeled by $2L^x + 2L^y + 2L^z - 3$ integers. Similarly, the gauge parameter $\hat{\alpha}^{k(ij)}$ can also have the above winding modes. Therefore, there are $N^{2L^x+2L^y+2L^z-3}$ winding modes that cannot be gauged away with their $W$ valued in $\mathbb{Z}_N$.

## 3.5 An Important Comment

When we studied the pure $\hat{\phi}$-theory (without gauge fields) in [2], we discussed configurations of the form

$$\hat{\phi}^{x(yz)} = -\hat{\phi}^{z(xy)} = 2\pi \left[ \frac{x}{\ell^x}\Theta(y - y_0) + \frac{y}{\ell^y}\Theta(x - x_0) - \frac{xy}{\ell^x\ell^y} \right]$$

$$\hat{\phi}^{y(xz)} = 0 \,. \tag{3.22}$$

The low-energy limit of the pure $\hat{\phi}$-theory has a global winding tensor symmetry. The winding charge of the configuration (3.22) is

$$Q^x = \frac{1}{2\pi} \oint dx \partial_x \hat{\phi}^{x(yz)} = \Theta(y - y_0). \tag{3.23}$$

Since it is not periodic in $y$, $Q^x$ is not a well-defined operator and therefore (3.22) violates the $U(1)$ winding tensor symmetry. However, since these configurations lead to states with energy of order $1/a$, while the states charged under that symmetry are at energy of order $a$, it is meaningful to ignore such configurations in the continuum limit. As a result, the continuum $\hat{\phi}$-theory exhibits the accidental $U(1)$ winding symmetry [2].

Let us turn now to the $\mathbb{Z}_N$ gauge theory. If instead of (3.10), we would have written the Lagrangian

$$\frac{\hat{\mu}_0}{12} \left( \partial_0 \hat{\phi}^{k(ij)} - N \hat{A}_0^{k(ij)} \right)^2 - \frac{\hat{\mu}}{2} \left( \partial_k \hat{\phi}^{k(ij)} - N \hat{A}^{ij} \right)^2 + \frac{1}{2\hat{g}_e^2} \hat{E}_{ij} \hat{E}^{ij} - \frac{1}{\hat{g}_m^2} \hat{B}^2 \,, \tag{3.24}$$

then the situation would have been as in the pure $\hat{\phi}$-theory. The configurations (3.22) would have been suppressed by their large action.

The Lagrangian (3.10) is the low energy limit of (3.24). Then, the equation of motion (3.12) relates (3.22) to

$$\hat{A}^{yz} = \frac{2\pi}{N} \left[ \frac{1}{\ell^x} \Theta(y - y_0) + \frac{y}{\ell^y} \delta(x - x_0) - \frac{y}{\ell^x \ell^y} \right] \tag{3.25}$$
$$\hat{A}^{zx} = \hat{A}^{xy} = 0 \,.$$

The lack of periodicity in $y$ means that we need a transition function at $y = \ell^y$,[6]

$$\hat{g}_{(y)}^{x(yz)} = -\hat{g}_{(y)}^{z(xy)} = \frac{2\pi}{N} \Theta(x - x_0) \,. \tag{3.26}$$

This transition function is inconsistent because $\exp(i\hat{g}_{(y)}^{x(yz)}) = \exp(-i\hat{g}_{(y)}^{z(xy)})$ is not periodic in $x$. Therefore, configurations like (3.22) are not allowed in the gauge theory.

The key point is the following. Considering only the space of fields, we can have configurations like (3.22) with trivial transition functions for the gauge field. (We do need transition functions for $\hat{\phi}$ when we view them as real fields with transition functions, rather than as circle-valued fields.) Then, the equation of motion (3.12), which is imposed as a constraint

---

[6]Viewed as real fields, the configuration (3.22) is not periodic in $x$. However, since it is periodic in $x$ as a circle-valued function and since the gauge field (3.25) is periodic in $x$, there is no need for a transition function at $x = \ell^x$.

in the continuum field theory, ties $\hat{\phi}$ to the gauge field and leads to the inconsistency.

We conclude that unlike the pure $\hat{\phi}$-theory in [2] or the gauge theory (3.24), where such configurations like (3.22) are allowed, but they are suppressed because of their action, here in (3.10) they are inconsistent with the gauge symmetry and must be excluded.

Even though configurations like (3.22) do not contribute, configurations of the form

$$
\begin{aligned}
&\hat{\phi}^{x(yz)} = -\hat{\phi}^{z(xy)} = 2\pi N \left[ \frac{x}{\ell^x} \Theta(y - y_0) + \frac{y}{\ell^y} \Theta(x - x_0) - \frac{xy}{\ell^x \ell^y} \right] \\
&\hat{\phi}^{y(xz)} = 0 \,,
\end{aligned}
\tag{3.27}
$$

are consistent with the gauge symmetry. Furthermore, unlike the case in [2], they are not suppressed by their action and they must be included.

Let us explore some properties of (3.27). First, although $Q^x$ of these configurations is ill-defined, $\exp(2\pi i Q^x/N)$ of (3.27) is well-defined. This means that these configurations respect the $\mathbb{Z}_N$ subgroup of the $U(1)$ winding tensor symmetry, which is the global symmetry of our gauge theory. In fact, for these configurations $\exp(2\pi i Q^x/N) = 1$, i.e. they are not charged under this $\mathbb{Z}_N$ global symmetry. Second, these configurations can be gauged away by choosing a large gauge transformation parameter $\hat{\alpha}^{k(ij)}$ of the form (3.22). Therefore, they do not contribute new states in addition to the $N^{2L^x+2L^y+2L^z-3}$ ground states we found above.

# 4 $BF$-type $\mathbb{Z}_N$ Tensor Gauge Theory

## 4.1 Continuum Lagrangian

We now discuss the third presentation of the $\mathbb{Z}_N$ tensor gauge theory [8].

This presentation involves the two gauge fields in Section 2 and 3:

$$
\begin{aligned}
(A_0, A_{ij}) : \quad & (\mathbf{1}, \mathbf{3}') \\
(\hat{A}_0^{i(jk)}, \hat{A}^{ij}) : \quad & (\mathbf{2}, \mathbf{3}')
\end{aligned}
\tag{4.1}
$$

where we have written their $S_4$ representations on the right. They are subject to two gauge transformations. The first one is

$$
\begin{aligned}
A_0 &\to A_0 + \partial_0 \alpha \,, \\
A_{ij} &\to A_{ij} + \partial_i \partial_j \alpha \,,
\end{aligned}
\tag{4.2}
$$

where $\alpha$ is a $2\pi$ periodic scalar. The second gauge transformation is

$$
\begin{aligned}
\hat{A}_0^{i(jk)} &\rightarrow \hat{A}_0^{i(jk)} + \partial_0 \hat{\alpha}^{i(jk)} \,, \\
\hat{A}^{ij} &\rightarrow \hat{A}^{ij} + \partial_k \hat{\alpha}^{k(ij)} \,,
\end{aligned}
\tag{4.3}
$$

where $\hat{\alpha}^{k(ij)}$ is $2\pi$ periodic and transforms in the **2** of $S_4$. Their electric and magnetic fields are given in (2.11) and (3.13). See [2] for details on the $A$ and $\hat{A}$ gauge fields.

The Euclidean Lagrangian is of the $BF$-type, i.e. it is a product of the gauge fields $(A_0, A_{ij})$ with the electric and magnetic fields $(\hat{E}^{ij}, \hat{B})$ for $\hat{A}$:

$$
\mathcal{L}_E = i\frac{N}{2\pi} \left( \frac{1}{2} A_{ij} \hat{E}^{ij} + A_0 \hat{B} \right) .
\tag{4.4}
$$

Integrating by parts, we can also write it as[7]

$$
\mathcal{L}_E = i\frac{N}{2\pi} \frac{1}{2} \left( \frac{1}{3} \hat{A}_0^{k(ij)} B_{k(ij)} - \hat{A}^{ij} E_{ij} \right) .
\tag{4.5}
$$

The equations of motion set all the gauge-invariant local operators to zero:

$$
E_{ij} = 0 \,, \quad B_{i(jk)} = 0 \,, \quad \hat{E}^{ij} = 0 \,, \quad \hat{B} = 0 \,.
\tag{4.6}
$$

While there are no local operators, there are gauge invariant extended operators, which generate exotic global symmetries. We will discuss this in detail in Section 4.2.

Since there are no local operators, the $\mathbb{Z}_N$ tensor gauge theory is robust (see [1] for a discussion of robustness). Small changes of the underlying microscopic model do not affect the long-distance field theory phase. In particular, the ground state degeneracy in Section 4.3 is also robust and cannot be lifted by small perturbations.

Similarly, as in the other two descriptions of the model, since there are no local operators, the possible universality violation due to higher-derivative terms, which was discussed in [1,2], is not present. Therefore the results computed from this continuum Lagrangian are universal.

*Quantization of the Level*

Let us now discuss the quantization of the coefficient $N$ in (4.4) and (4.5). Similar to the ordinary $BF$-theories, the coefficient here will be quantized by large gauge transformation in the presence of nontrivial fluxes.

---

[7]We have used $(\partial_i A_{jk} + \partial_j A_{ki} + \partial_k A_{ij})\hat{A}_0^{k(ij)} = 0$. Also note that $B_{k(ij)} = B_{[ki]j} + B_{[kj]i} = 2\partial_k A_{ij} - \partial_i A_{kj} - \partial_j A_{ki}$.

Consider the following large gauge transformation on a Euclidean 4-torus:

$$\alpha = 2\pi \frac{\tau}{\ell^\tau} \,. \tag{4.7}$$

Under this gauge transformation, the action from (4.4) changes by

$$iN \oint dx dy dz \hat{B} \,. \tag{4.8}$$

From [2], we have the following quantized fluxes

$$\hat{b}^x \equiv \int_{x_1}^{x_2} dx \oint dy \oint dz \hat{B} \in 2\pi\mathbb{Z} \,, \tag{4.9}$$

and similar fluxes for the other directions. Therefore for the path integral to be invariant under this large gauge transformation, we need

$$N \in \mathbb{Z} \,. \tag{4.10}$$

For completeness, let us record the other quantized fluxes from [2] below:

$$e_{xy}(x_1, x_2) \equiv \oint d\tau \int_{x_1}^{x_2} dx \oint dy E_{xy} \in 2\pi\mathbb{Z} \,. \tag{4.11}$$

$$b_{[yz]x}(x_1, x_2) \equiv \int_{x_1}^{x_2} dx \oint dy \oint dz \, B_{[yz]x} \in 2\pi\mathbb{Z} \,. \tag{4.12}$$

$$\hat{e}^{xy}(x, y) \equiv \oint d\tau \oint dz \, \hat{E}^{xy} \in 2\pi\mathbb{Z} \,. \tag{4.13}$$

## 4.2   Defects and Operators

While there are no gauge-invariant local operators in the $\mathbb{Z}_N$ tensor gauge theory, there are gauge-invariant non-local, extended observables analogous to the Wilson lines in the (2+1)d Chern-Simons theory.

*Fractons and Planons as Defects of A*

The simplest defect is a single particle of gauge charge +1 at a fixed point in space

$(x, y, z)$. It is captured by the gauge-invariant defect

$$\exp\left[i \int_{-\infty}^{\infty} dt\, A_0(t, x, y, z)\right] . \tag{4.14}$$

This immobile particle is identified as the probe limit of a fracton. The gauge charge is quantized by the large gauge transformation (4.7).

A pair of fractons of gauge charges $\pm 1$ separated, say, in the $z$-direction, can move collectively. This is captured by the defect:

$$W(z_1, z_2, \mathcal{C}) = \exp\left[i \int_{z_1}^{z_2} dz \int_{\mathcal{C}} (\, dt\partial_z A_0 + dx A_{xz} + dy A_{yz}\,)\right] . \tag{4.15}$$

where $\mathcal{C}$ is a spacetime curve in $(t, x, y)$ (but no $z$) representing the motion of a dipole of fractons on the $(x, y)$-plane. It is a planon on the $(x, y)$-plane.

### Lineons and Planons as Defects of $\hat{A}$

The second kind of particle has three variants each associated with a spatial direction. A static particle of species $x^i$ and gauge charge $+1$ is captured by the following defect

$$\exp\left[i \int_{-\infty}^{\infty} dt \hat{A}_0^{i(jk)}\right] . \tag{4.16}$$

The gauge charge is quantized by a large gauge transformation $\hat{\alpha}^{i(jk)} = -\hat{\alpha}^{j(ki)} = 2\pi\frac{\tau}{\ell^\tau}, \hat{\alpha}^{k(ij)} = 0$. The particle of species, say, $z$ moving in the $z$-direction is captured by the following line defect in spacetime

$$\hat{W}^z(x, y, \hat{\mathcal{C}}) = \exp\left[i \int_{\hat{\mathcal{C}}} \left(\hat{A}_0^{z(xy)} dt + \hat{A}^{xy} dz\right)\right] , \tag{4.17}$$

where $\hat{\mathcal{C}}$ is a spacetime curve on the $(t, z)$-plane representing the motion of a particle along the $z$-direction. The particle by itself cannot turn in space; it is confined to move along the $z$-direction. This particle is the probe limit of the lineon.

While a single lineon of species $x^i$ is confined to move along the $x^i$ direction, a pair of them can move in more general directions. For example, a pair of lineons of species $x$ separated in the $z$ direction can move on the $xy$-plane. This motion is captured by the defect

$$\hat{P}(z_1, z_2, \mathcal{C}) = \exp\left[i \int_{z_1}^{z_2} dz \int_{\mathcal{C}} \left(\partial_z \hat{A}_0^{x(yz)} dt + \partial_z \hat{A}^{yz} dx - \partial_z \hat{A}^{zx} dy - \partial_y \hat{A}^{xy} dy\right)\right] \tag{4.18}$$

where $\mathcal{C}$ is a spacetime curve in $(t, x, y)$ representing a dipole of lineons, i.e. a planon, on the $(x, y)$-plane.

## *Quasi-Topological Defects*

If we deform infinitesimally the spacetime curve $\mathcal{C}$ to a nearby one $\mathcal{C}'$ and similarly, the spacetime curve $\hat{\mathcal{C}}$ to a nearby one $\hat{\mathcal{C}}'$, the changes in these defects can be computed using the Stokes theorem:

$$
\begin{aligned}
\frac{W(z_1, z_2, \mathcal{C})}{W(z_1, z_2, \mathcal{C}')} &= \exp\left[i \int_{z_1}^{z_2} dz \int_{\mathcal{S}} \left(E_{zx} dt dx - E_{zy} dy dt + B_{[xy]z} dx dy\right)\right], \\
\frac{\hat{W}^z(x, y, \hat{\mathcal{C}})}{\hat{W}^z(x, y, \hat{\mathcal{C}}')} &= \exp\left[i \int_{\hat{\mathcal{S}}} \hat{E}^{xy} dt dz\right], \\
\frac{\hat{P}(z_1, z_2, \mathcal{C})}{\hat{P}(z_1, z_2, \mathcal{C}')} &= \exp\left[i \int_{z_1}^{z_2} dz \int_{\mathcal{S}} \left(\partial_z \hat{E}^{yz} dt dx + \partial_z \hat{E}^{xz} dy dt + \partial_y \hat{E}^{xy} dy dt - \hat{B} dx dy\right)\right],
\end{aligned}
\tag{4.19}
$$

where $\mathcal{S}$ is a surface bounded by $\mathcal{C}$ and $\mathcal{C}'$ and $\hat{\mathcal{S}}$ is a surface bounded by $\hat{\mathcal{C}}$ and $\hat{\mathcal{C}}'$. In the $\mathbb{Z}_N$ tensor gauge theory, the equations of motion set $E_{ij} = B_{[ij]k} = \hat{E}^{ij} = \hat{B} = 0$, so these defects are invariant under small deformations of $\mathcal{C}$ and $\hat{\mathcal{C}}$ in the appropriate manifold. Similar properties are true for defects along the other directions.

To conclude, these defects are topological under deformations along certain directions, but not all.

## *Symmetry Operators*

In the special case when $\mathcal{C}$ is a space-like curve on the $xy$-plane, $W(z_1, z_2, \mathcal{C})$ reduces to the $\mathbb{Z}_N$ dipole symmetry operator (2.12). Similarly, in the special case when $\hat{\mathcal{C}}$ is a line along the $z$ direction at a fixed time, $\hat{W}^z(x, y, \hat{\mathcal{C}})$ reduces to the $\mathbb{Z}_N$ tensor symmetry operator $\hat{W}^z(x, y)$ (3.14).

When the two $\mathbb{Z}_N$ symmetry operators $W(z_1, z_2, \mathcal{C})$ and $\hat{W}^x(y_0, z_0)$ act at the same time with $\mathcal{C}$ a curve in the $xy$-plane, they obey the commutation relation

$$
\hat{W}^x(y_0, z_0) W(z_1, z_2, \mathcal{C}) = e^{2\pi i I(\mathcal{C}, y_0)/N} W(z_1, z_2, \mathcal{C}) \hat{W}^x(y_0, z_0), \quad \text{if } z_1 < z_0 < z_2. \tag{4.20}
$$

Here $I(\mathcal{C}, y_0)$ is the intersection number between the curve $\mathcal{C}$ and the $y = y_0$ line on the $xy$-plane.[8] There are similar commutation relations for operators in the other directions.

---

[8]At the risk of confusing the reader, we would like to point out that this lack of commutativity can be

Next, consider a planon $\hat{P}(x_1, x_2, \mathcal{C})$, say, separated in the $x$ direction with $\mathcal{C}$ a spacetime curve in $(t, y, z)$. In the special case when $\mathcal{C}$ is a closed line along the $z$ direction at a fixed time and $y = y_0$, $\hat{P}(x_1, x_2, \mathcal{C})$ reduces to a pair of $\mathbb{Z}_N$ tensor symmetry operators $\hat{W}^z(x_1, y_0)^{-1}\hat{W}^z(x_2, y_0)$. The invariance (4.19) under small deformation of $\mathcal{C}$ implies that $\hat{W}^z(x_1, y_0)^{-1}\hat{W}^z(x_2, y_0)$ is independent of $y_0$. Indeed, this follows from (3.18) which we have discussed before.

## 4.3    Ground State Degeneracy

We now study the ground states of the $\mathbb{Z}_N$ tensor gauge theory on a spatial 3-torus $\mathbb{T}^3$ using the presentation (4.4). The discussion will be similar to that in [8] and will extend it by paying attention to the global issues and the precise space of fields. The analysis proceeds similarly as the $2 + 1$-dimensional $\mathbb{Z}_N$ tensor gauge theory in Section 7.5 of [1].

Let us choose the temporal gauge setting $A_0 = 0$ and $\hat{A}_0^{i(jk)} = 0$. Then the phase space is

$$\left\{ A_{ij}, \hat{A}^{ij} \;\middle|\; B_{[ij]k} = 0, \;\hat{B} = 0 \right\} \Big/ \left\{ A_{ij} \sim A_{ij} + \partial_i \partial_j \alpha, \;\hat{A}^{ij} \sim \hat{A}^{ij} + \partial_k \hat{\alpha}^{k(ij)} \right\}, \qquad (4.21)$$

where we mod out the time-independent gauge transformations. The solution modulo gauge transformations is

$$A_{ij} = \frac{1}{\ell^j} f_{ij}^i(x^i) + \frac{1}{\ell^i} f_{ij}^j(x^j) \,,$$
$$\hat{A}^{ij} = \frac{1}{\ell^k} \hat{f}_i^{ij}(x^i) + \frac{1}{\ell^k} \hat{f}_j^{ij}(x^j) \,. \qquad (4.22)$$

We have put in factors of $\ell^i$ for later convenience. The functions $f_{ij}^i$ have mass dimension 1 while $\hat{f}_i^{ij}$ are dimensionless.

Only the sum of the zero modes for $\frac{1}{\ell^k}\hat{f}_i^{ij}(x^i) + \frac{1}{\ell^k}\hat{f}_j^{ij}(x^j)$ is physical, and similarly for $f$. This implies a gauge symmetry for $\hat{f}$:

$$\hat{f}_i^{ij}(t, x^i) \rightarrow \hat{f}_i^{ij}(t, x^i) + c(t) \,,$$
$$\hat{f}_j^{ij}(t, x^j) \rightarrow \hat{f}_j^{ij}(t, x^j) - c(t) \,. \qquad (4.23)$$

There is a similar gauge symmetry for $f$. As in [1], we define the gauge-invariant modes

$$\bar{f}_{ij}^i(t, x^i) = f_{ij}^i(t, x^i) + \frac{1}{\ell^i} \oint dx^j f_{ij}^j(t, x^j) \,. \qquad (4.24)$$

interpreted as a mixed anomaly between these two $\mathbb{Z}_N$ symmetries. See [29] for a related discussion on the relativistic one-form symmetries in the $2 + 1$-dimensional $\mathbb{Z}_N$ gauge theory.

They are subject to the constraint

$$\oint dx^i \, \bar{f}^i_{ij}(t, x^i) = \oint dx^j \, \bar{f}^j_{ij}(t, x^j) \,. \tag{4.25}$$

Let us discuss the global periodicities of $\hat{f}$ and $\bar{f}$. The large gauge transformation $\hat{\alpha}^{z(xy)} = -\hat{\alpha}^{y(zx)} = 2\pi \frac{z}{\ell^z} w(x), \hat{\alpha}^{x(yz)} = 0$, with $w(x)$ a piecewise continuous integer-valued function, implies that $\hat{f}$ has a point-wise $2\pi$ periodicity:

$$\begin{aligned}
\hat{f}^{xy}_x(x) &\to \hat{f}^{xy}_x(x) + 2\pi w(x) \,, \\
\hat{f}^{xy}_y(y) &\to \hat{f}^{xy}_y(y) \,,
\end{aligned} \tag{4.26}$$

and similarly for the $y$ direction and for the other components of $\hat{f}^{ij}_i$.

On the other hand, the large gauge transformation

$$\alpha = 2\pi \left[ \frac{x}{\ell^x} \Theta(x - x_0) + \frac{y}{\ell^y} \Theta(y - y_0) - \frac{xy}{\ell^x \ell^y} \right] \tag{4.27}$$

implies that $\hat{f}$ has a point-wise delta function periodicity:

$$\begin{aligned}
\bar{f}^x_{xy}(t, x) &\to \bar{f}^x_{xy}(t, x) + 2\pi \delta(x - x_0) \,, \\
\bar{f}^y_{xy}(t, y) &\to \bar{f}^y_{xy}(t, y) + 2\pi \delta(y) \,,
\end{aligned} \tag{4.28}$$

for each $x_0$, and

$$\begin{aligned}
\bar{f}^x_{xy}(t, x) &\to \bar{f}^x_{xy}(t, x) \,, \\
\bar{f}^y_{xy}(t, y) &\to \bar{f}^y_{xy}(t, y) + 2\pi \delta(y - y_0) - 2\pi \delta(y) \,,
\end{aligned} \tag{4.29}$$

for each $y_0$. The other components of $f^i_{ij}$ have similar periodicity.

The effective Lagrangian written in terms of $\bar{f}$ and $\hat{f}$ is

$$L_{eff} = \frac{N}{2\pi} \sum_{i<j} \left[ \oint dx^i \, \hat{f}^{ij}_i(t, x^i) \partial_0 \bar{f}^i_{ij}(t, x^i) + \oint dx^j \, \hat{f}^{ij}_j(t, x^j) \partial_0 \bar{f}^j_{ij}(t, x^j) \right] \,. \tag{4.30}$$

The Lagrangian for these modes is effectively $1 + 1$-dimensional. In the strict continuum limit, the ground state degeneracy is infinite.

Let us regularize the degeneracy by placing the theory on a lattice. We will focus on the modes $\hat{f}^{xy}_i$ and $\bar{f}^i_{xy}$, while the other modes can be done in parallel. On a lattice, we can solve $\bar{f}^y_{xy}(\hat{y} = L^y)$ in terms of $\bar{f}^x_{xy}(\hat{x})$ and the other $\bar{f}^y_{xy}(\hat{y})$ using (4.25). The remaining, unconstrained $L^x + L^y - 1$ $\bar{f}$'s have periodicities $\bar{f}^i_{xy}(\hat{x}^i) \sim \bar{f}^i_{xy}(\hat{x}^i) + 2\pi/a$ for each $\hat{x}^i$. On the other hand, we can use the gauge symmetry (4.23) to gauge fix $\hat{f}^{xy}_y(\hat{y} = L^y) = 0$. The

remaining $L^x + L^y - 1$ $\hat{f}^{xy}$'s have periodicities $\hat{f}_i^{xy}(\hat{x}^i) \sim \hat{f}_i^{xy}(\hat{x}^i) + 2\pi$ for each $\hat{x}^i$. The effective Lagrangian is now written in terms of $L^x + L^y - 1$ pairs of variables $\left( \hat{f}_i^{xy}(\hat{x}^i), \bar{f}_{xy}^i(\hat{x}^i) \right)$.

Each pair $\left( \hat{f}_i^{xy}(\hat{x}^i), \bar{f}_{xy}^i(\hat{x}^i) \right)$ leads to an $N$-dimensional Hilbert space. Combining the modes from the other directions, we end up with the expected ground state degeneracy $N^{2L^x+2L^y+2L^z-3}$.

*Ground State Degeneracy from Global Symmetries*

The ground state degeneracy can be understood from the $\mathbb{Z}_N$ global symmetries. Let us

| lattice model | $(2+1)d$ toric code | $(3+1)d$ X-cube model |
|---|---|---|
| excitations | anyons | fractons, lineons, planons |
| ground state degeneracy on a torus | $N^2$ | $N^{2L^x+2L^y+2L^z-3}$ |
| continuum field theory | $\mathbb{Z}_N$ gauge theory | $\mathbb{Z}_N$ tensor gauge theory |
| Lagrangian | $i\frac{N}{2\pi}\hat{A}dA$ | $i\frac{N}{2\pi}\left( \frac{1}{2}A_{ij}\hat{E}^{ij} + A_0\hat{B} \right)$ |
| gauge fields | $A_\mu \to A_\mu + \partial_\mu\alpha$<br>$\hat{A}_\mu \to \hat{A}_\mu + \partial_\mu\hat{\alpha}$ | $A_0 \to A_0 + \partial_0\alpha$<br>$A_{ij} \to A_{ij} + \partial_i\partial_j\alpha$<br>$\hat{A}_0^{i(jk)} \to \hat{A}_0^{i(jk)} + \partial_0\hat{\alpha}^{i(jk)}$<br>$\hat{A}^{ij} \to \hat{A}^{ij} + \partial_k\hat{\alpha}^{k(ij)}$ |
| EoM | $dA = d\hat{A} = 0$ | $E_{ij} = B_{[ij]k} = \hat{E}^{ij} = \hat{B} = 0$ |
| defect | Wilson line<br>$\exp\left[ in \oint A + im \oint \hat{A} \right]$ | Wilson line/strip<br>$W(x_1^k, x_2^k, \mathcal{C})$<br>$\hat{W}^k(x^i, x^j, \hat{\mathcal{C}})$<br>$\hat{P}(x_1^k, x_2^k, \mathcal{C})$ |

Table 4: Analogy between the toric code and the X-cube model. Here $\mathcal{C}$ is a spacetime curve in $(t, x^i, x^j)$ and $\hat{\mathcal{C}}$ is a spacetime curve in $(t, x^k)$.

focus on a subset of the symmetry operators: the $\mathbb{Z}_N$ tensor symmetry operator extended along the $z$ direction (3.14)

$$\hat{W}^z(x_0, y_0) \tag{4.31}$$

and the $\mathbb{Z}_N$ dipole symmetry strip operators on the $xy$-plane (2.12)

$$W(x_1, x_2, \mathcal{C}_y^{yz}), \quad W(y_1, y_2, \mathcal{C}_x^{xz}). \tag{4.32}$$

Here $\mathcal{C}_x^{xy}$ is a curve on the $xy$-plane that wraps around $x$ direction once but not the $y$ direction and similarly with $x \leftrightarrow y$. Due to the topological property (4.19), these strip operators on the $xy$-plane do not depend on their $z$ coordinates. These operators obey commutation relations similar to (4.20):

$$\begin{aligned}
\hat{W}^z(x_0, y_0) W(x_1, x_2, \mathcal{C}_y^{yz}) &= e^{2\pi i/N} W(x_1, x_2, \mathcal{C}_y^{yz}) \hat{W}^z(x_0, y_0), \quad \text{if } x_1 < x_0 < x_2, \\
\hat{W}^z(x_0, y_0) W(y_1, y_2, \mathcal{C}_x^{xz}) &= e^{2\pi i/N} W(y_1, y_2, \mathcal{C}_x^{xz}) \hat{W}^z(x_0, y_0), \quad \text{if } y_1 < y_0 < y_2,
\end{aligned} \tag{4.33}$$

and they commute otherwise.

On a lattice, due to (3.18), we have $L^x + L^y - 1$ tensor symmetry operators (4.31) along the $z$ direction. Similarly, due to (2.15), we have $L^x + L^y - 1$ dipole symmetry operators (4.32) on the $xy$ plane. The commutation relations between these operators are isomorphic to $L^x + L^y - 1$ copies of the $\mathbb{Z}_N$ Heisenberg algebra, $AB = e^{2\pi i/N} BA$ and $A^N = B^N = 1$. The isomorphism is given by

$$\begin{aligned}
A_{\hat{x}} &= \hat{W}^z(\hat{x}, 1), & B_{\hat{x}} &= W_{(x)}(\hat{x}), & \hat{x} &= 1, \cdots, L^x, \\
A_{\hat{y}} &= \hat{W}^z(1, \hat{y}) \hat{W}^z(1, 1)^{-1}, & B_{\hat{y}} &= W_{(y)}(\hat{y}), & \hat{y} &= 2, \cdots, L^y,
\end{aligned} \tag{4.34}$$

where $W_{(x)}(\hat{x}) \equiv \exp\left[ia \oint dy A_{xy}\right]$ is a strip operator along the $y$ direction with width $a$, and similarly for $W_{(y)}(\hat{y})$.

Combining the symmetry operators from the other directions, the commutation relations force the ground state degeneracy to be $N^{2L^x + 2L^y + 2L^z - 3}$.[9]

Finally, we summarize the analogy between the toric code and the X-cube model (which will be discussed in Section 5.2) as well as their continuum limits in Table 4.

# 5  Dualities

In this section we discuss the dualities of our continuum and lattice theories.

---

[9] For ordinary $2+1$-dimensional $\mathbb{Z}_N$ gauge theory on a 2-torus, the electric and magnetic one-form global symmetries give rise to 2 pairs of $\mathbb{Z}_N$ Heisenberg algebra. Hence the ground state degeneracy is $N^2$.

In Section 5.1, we show that the continuum $\mathbb{Z}_N$ theory of $A$ (Section 2.2), that of $\hat{A}$ (Section 3.2), and the $BF$-type theory (Section 4.1) are dual to each other. These are exact dualities of continuum field theories.

In Section 5.2, we show that the lattice $\mathbb{Z}_N$ theory of $A$ (Section 2.1), that of $\hat{A}$ (Section 3.1), and the X-cube model are dual to each other at long distances. These are infrared dualities. The low energy limit is the continuum $\mathbb{Z}_N$ tensor gauge theory. We further discuss the global symmetries of these lattice models.

## 5.1 The Three Continuum Descriptions

We now show the equivalence between the three different presentations, (2.8), (3.10), and (4.4), of the $\mathbb{Z}_N$ tensor gauge theory by duality transformations.

We first show that (2.8) and (4.4) are dual to each other. We start with (2.8)

$$\mathcal{L}_E = -\frac{i}{2(2\pi)}\hat{E}^{ij}(\partial_i\partial_j\phi - NA_{ij}) - \frac{i}{2\pi}\hat{B}(\partial_0\phi - NA_0)\,, \tag{5.1}$$

where $(A_0, A_{ij})$ are the $U(1)$ tensor gauge fields and $\phi$ is a $2\pi$-periodic real scalar field that Higgses the $U(1)$ gauge symmetry to $\mathbb{Z}_N$. The fields $\hat{E}^{ij}$ in the $\mathbf{3'}$ and $\hat{B}$ in the $\mathbf{1}$ are the Lagrangian multipliers.

We rewrite the Lagrangian as

$$\mathcal{L}_E = i\frac{N}{2\pi}\left(\frac{1}{2}A_{ij}\hat{E}^{ij} + A_0\hat{B}\right) + i\frac{\phi}{2\pi}\left(-\frac{1}{2}\partial_i\partial_j\hat{E}^{ij} + \partial_0\hat{B}\right)\,. \tag{5.2}$$

Now we interpret the Higgs field $\phi$ as a Lagrangian multiplier implementing the constraint

$$\frac{1}{2}\partial_i\partial_j\hat{E}^{ij} = \partial_0\hat{B}\,. \tag{5.3}$$

Locally, the constraint is solved by gauge fields $(\hat{A}_0^{i(jk)}, \hat{A}^{ij})$ in the $(\mathbf{2}, \mathbf{3'})$:

$$\begin{aligned}\hat{E}^{ij} &= \partial_0\hat{A}^{ij} - \partial_k\hat{A}_0^{k(ij)}\\\hat{B} &= \frac{1}{2}\partial_i\partial_j\hat{A}^{ij}\,.\end{aligned} \tag{5.4}$$

(5.2) then reduces to (4.4). Hence we have shown the equivalence between (2.8) and (4.4).

Next we show that (3.10) is dual to (4.4). We start with (3.10):

$$\mathcal{L}_E = \frac{i}{2(2\pi)} E_{ij} \left( \partial_k \hat{\phi}^{k(ij)} - N \hat{A}^{ij} \right) - \frac{i}{6(2\pi)} B_{k(ij)} \left( \partial_0 \hat{\phi}^{k(ij)} - N \hat{A}_0^{k(ij)} \right) \tag{5.5}$$

where $(\hat{A}_0^{k(ij)}, \hat{A}^{ij})$ are gauge fields in the $(\mathbf{2}, \mathbf{3'})$ of $S_4$ and $\hat{\phi}^{k(ij)}$ is a Higgs field in the $\mathbf{2}$ with charge $N$. $E_{ij}$ and $B_{k(ij)}$ are Lagrangian multipliers in the $\mathbf{3'}$ and $\mathbf{2}$ of $S_4$, respectively. We can rewrite the Lagrangian as

$$\mathcal{L}_E = i \frac{N}{2\pi} \left( -\frac{1}{2} \hat{A}^{ij} E_{ij} + \frac{1}{6} \hat{A}^{k(ij)} B_{k(ij)} \right) - \frac{i}{6(2\pi)} \hat{\phi}^{k(ij)} \left( 2\partial_k E_{ij} - \partial_i E_{jk} - \partial_j E_{ki} - \partial_0 B_{k(ij)} \right) \tag{5.6}$$

where we have used $\hat{\phi}^{k(ij)}(\partial_k E_{ij} + \partial_i E_{jk} + \partial_j E_{ki}) = 0$. We now interpret the Higgs field $\hat{\phi}^{k(ij)}$ as a Lagrangian multiplier implementing the constraint

$$2\partial_k E_{ij} - \partial_i E_{jk} - \partial_j E_{ki} = \partial_0 B_{k(ij)} \,. \tag{5.7}$$

This constraint can be locally solved by gauge fields $(A_0, A_{ij})$ in the $(\mathbf{1}, \mathbf{3'})$:

$$\begin{aligned} E_{ij} &= \partial_0 A_{ij} - \partial_i \partial_j A_0 \,, \\ B_{k(ij)} &= 2\partial_k A_{ij} - \partial_i A_{kj} - \partial_j A_{ki} \,. \end{aligned} \tag{5.8}$$

(5.6) then reduces to (4.5). Finally, we integrate (4.5) by parts to arrive at (4.4).

We conclude that the Lagrangians (2.8), (3.10), and (4.4) are three different presentations of the same continuum field theory.

## 5.2   X-Cube Model and the $\mathbb{Z}_N$ Lattice Tensor Gauge Theories

In this subsection we realize the X-cube model as limits of the $\mathbb{Z}_N$ lattice gauge theories of $A$ (Section 2.1) and $\hat{A}$ (Section 3.1). We further discuss the global symmetries of these three lattice theories.

### $\mathbb{Z}_N$ Lattice Gauge Theory of $A$

As discussed in Section 2.1, the $\mathbb{Z}_N$ lattice $A$ theory has an electric tensor global symmetry. Its conserved symmetry operator is a line in the, say, $z$ direction and at a fixed point $(\hat{x}_0, \hat{y}_0)$ (2.6):

$$\mathcal{U}^z(\hat{x}_0, \hat{y}_0) \sim \prod_{\hat{z}=1}^{L^z} V_{xy}(\hat{x}_0, \hat{y}_0, \hat{z}) \,. \tag{5.9}$$

Gauss law (2.4), which is strictly imposed, implies that the $\hat{x}_0, \hat{y}_0$ dependence of $\mathcal{U}^z(\hat{x}_0, \hat{y}_0)$ factorizes:

$$\mathcal{U}^z(\hat{x}_0, \hat{y}_0) = \mathcal{U}^z_x(\hat{x}_0)\,\mathcal{U}^z_y(\hat{y}_0)\,. \tag{5.10}$$

Similarly, there are conserved operators along the other directions. These are recognized as the symmetry operators $(\mathbf{3}', \mathbf{2})$ tensor global symmetry in Appendix B.1.

Note that the $(\mathbf{3}', \mathbf{1})$ dipole global symmetry in the continuum (2.2) is not present on the lattice.

## $\mathbb{Z}_N$ Lattice Gauge Theory of $\hat{A}$

As discussed in Section 3.1, the $\mathbb{Z}_N$ lattice $\hat{A}$ theory has an electric dipole global symmetry. Its conserved symmetry operator is proportional to

$$\mathcal{U}(\hat{z}, \mathcal{C}^{xy}) \sim \prod_{\mathcal{C}^{xy}} \hat{V} \tag{5.11}$$

where the product is over a zigzagging closed curve $\mathcal{C}^{xy}$ on the $xy$ plane (see Figure 6). Special cases of such strip operators are in (3.7).

Gauss law (3.5), which is strictly imposed, implies that $\mathcal{U}(\hat{z}, \mathcal{C}^{xy})$ is invariant under small changes of $\mathcal{C}^{xy}$. In other words, the dependence of $\mathcal{U}(\hat{z}, \mathcal{C}^{xy})$ on $\mathcal{C}^{xy}$ is topological. Similarly, there are conserved operators along the other directions. These are recognized as the symmetry operators for the $(\mathbf{3}', \mathbf{1})$ tensor global symmetry in Appendix B.2.[10]

Note that the $(\mathbf{3}', \mathbf{2})$ tensor global symmetry in the continuum (3.2) is not present on the lattice.

## X-Cube Model

The X-cube model [33] can be realized as the limit $\hat{g}_e \to \infty$ of the $\mathbb{Z}_N$ lattice gauge theory of $\hat{A}$ (3.8)

$$H = -\frac{1}{\hat{g}_m^2} \sum_{\text{cubes}} \hat{L} - \frac{1}{\hat{g}} \sum_{\text{sites}} (\hat{G}^{x(yz)} + \hat{G}^{y(zx)} + \hat{G}^{z(xy)}) + c.c. \tag{5.12}$$

where the individual terms are defined in Section 3.1 and Figure 4. Note that there are no gauge symmetry or Gauss law in the X-cube model.

---

[10]In the continuum limit, $\mathcal{U}(\hat{z}, \mathcal{C}^{xy})$ is identified with $\mathcal{U}(z, z+a, \mathcal{C}^{xy})$ in Appendix B.2 where $a$ is the lattice spacing.

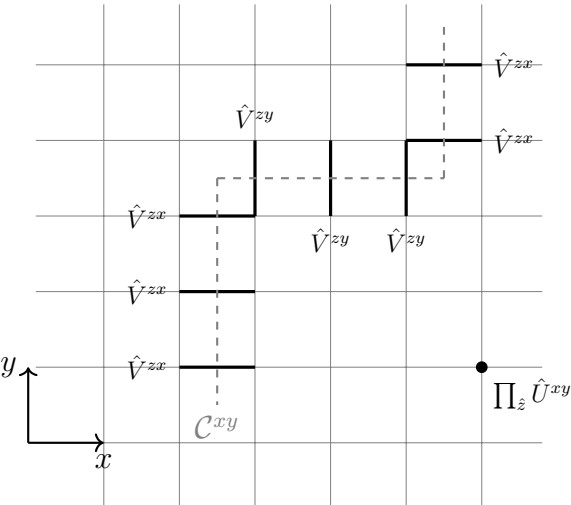

Figure 6: The symmetry operator $\mathcal{U}^z(\hat{x}_0, \hat{y}_0)$ of the $(\mathbf{3'}, \mathbf{2})$ (unconstrained) tensor symmetry is a product of $\hat{U}^{xy}$ along the $z$ direction (not shown in the above figure) at a fixed point $(\hat{x}_0, \hat{y}_0)$ on the $xy$-plane. The symmetry operator $\mathcal{U}(\hat{z}, \mathcal{C}^{xy})$ of the $(\mathbf{3'}, \mathbf{1})$ (unconstrained) dipole symmetry is a product of $\hat{V}^{zx}$ and $\hat{V}^{zy}$ along a closed curve $\mathcal{C}^{xy}$ on the $xy$-plane at a fixed $\hat{z}$.

The X-cube model has two kinds of global symmetries. The conserved symmetry operator of the first kind is the Wilson line operator (3.4)

$$\mathcal{U}^z(\hat{x}_0, \hat{y}_0) \sim \prod_{\hat{z}=1}^{L^z} \hat{U}^{xy}(\hat{x}_0, \hat{y}_0, \hat{z}) \,. \tag{5.13}$$

Similarly there are other line operators along the other directions. These are the string-like logical operators of the X-cube model.

Unlike the symmetry operator (5.9) in the $\mathbb{Z}_N$ lattice gauge theory of $A$, the $\hat{x}_0, \hat{y}_0$ dependence of (5.13) does not factorize as in (5.10). It is the unconstrained $(\mathbf{3'}, \mathbf{2})$ tensor global symmetry in Appendix B.1.

The conserved symmetry operator of the second kind is (5.11). However, in the X-cube model, the operator $\mathcal{U}(\hat{z}, \mathcal{C}^{xy})$ depends not only on the topology of the curve $\mathcal{C}^{xy}$, but also the detailed shape of it. Similarly there are other line operators along the other directions. These are the membrane-like logical operators of the X-cube model. It is the unconstrained $(\mathbf{3'}, \mathbf{1})$ dipole global symmetry in Appendix B.2.

Dually, the X-cube model can also be realized as the limit $g_e \to \infty$ of the $\mathbb{Z}_N$ lattice gauge theory of $A$ (2.7) on the dual lattice. In this presentation, the unconstrained $(\mathbf{3'}, \mathbf{2})$ tensor symmetry is the electric symmetry (2.6). On the other hand, the symmetry operator

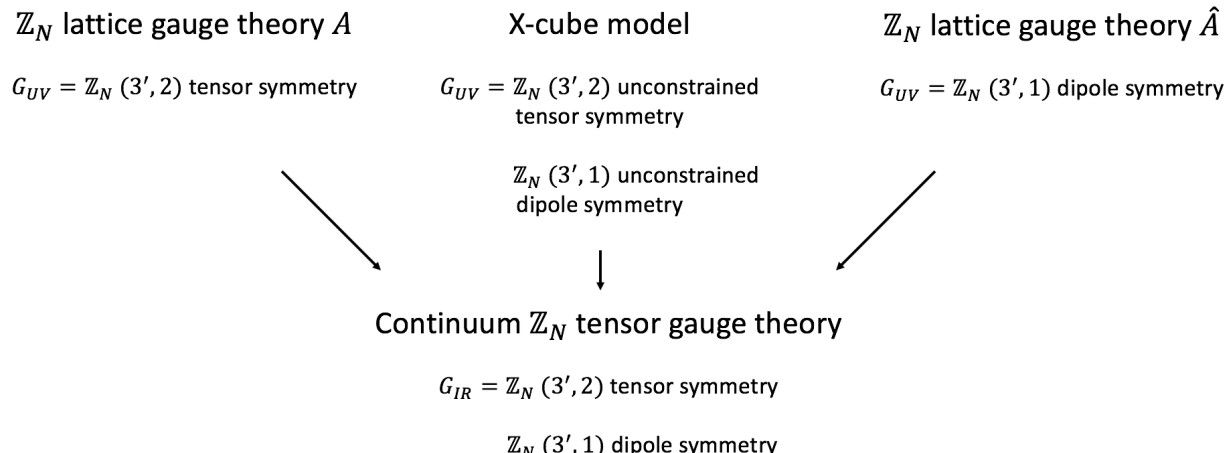

Figure 7: The three lattice models, the $\mathbb{Z}_N$ $A$-theory, the $\mathbb{Z}_N$ $\hat{A}$-theory, and the X-cube model are dual to each other at long distances. The low energy continuum field theory is the $\mathbb{Z}_N$ tensor gauge theory. We also show the microscopic global symmetry $G_{UV}$ of each lattice model, and the emergent global symmetry $G_{IR}$ at long distances.

of the unconstrained $(\mathbf{3}', \mathbf{1})$ dipole symmetry is the Wilson strip (2.3).

These unconstrained tensor and dipole symmetries are analogous to the non-relativistic electric and magnetic one-form symmetries of the toric code [6]. At long distances, they become the tensor and dipole symmetries of the continuum $\mathbb{Z}_N$ tensor gauge theory (see Section 2.3, 3.3, and 4.2).

We conclude that the $\mathbb{Z}_N$ lattice gauge theory of $A$, that of $\hat{A}$, and the X-cube model are dual to each other at long distances. We summarize their global symmetries on the lattice and in the continuum in Figure 7.

# Acknowledgements

We thank X. Chen, M. Cheng, M. Fisher, A. Gromov, M. Hermele, P.-S. Hsin, A. Kitaev, D. Radicevic, L. Radzihovsky, S. Sachdev, D. Simmons-Duffin, S. Shenker, K. Slagle, D. Stanford for helpful discussions. We also thank P. Gorantla, H.T. Lam, D. Radicevic, and T. Rudelius for comments on the manuscript. The work of N.S. was supported in part by DOE grant DE$-$SC0009988. NS and SHS were also supported by the Simons Collaboration on Ultra-Quantum Matter, which is a grant from the Simons Foundation (651440, NS). Opinions and conclusions expressed here are those of the authors and do not necessarily reflect the views of funding agencies.

# A  Cubic Group and Our Notations

The symmetry group of the cubic lattice (up to translations) is the *cubic group*, which consists of 48 elements. We will focus on the group of orientation-preserving symmetries of the cube, which is isomorphic to the permutation group of four objects $S_4$.

The irreducible representations of $S_4$ are the trivial representation $\mathbf{1}$, the sign representation $\mathbf{1'}$, a two-dimensional irreducible representation $\mathbf{2}$, the standard representation $\mathbf{3}$, and another three-dimensional irreducible representation $\mathbf{3'}$. $\mathbf{3'}$ is the tensor product of the sign representation and the standard representation, $\mathbf{3'} = \mathbf{1'} \otimes \mathbf{3}$.

It is convenient to embed $S_4 \subset SO(3)$ and decompose the known $SO(3)$ irreducible representations in terms of $S_4$ representations. The first few are

$$
\begin{array}{ccl}
SO(3) & \supset & S_4 \\
\mathbf{1} & = & \mathbf{1} \\
\mathbf{3} & = & \mathbf{3} \\
\mathbf{5} & = & \mathbf{2} \oplus \mathbf{3'} \\
\mathbf{7} & = & \mathbf{1'} \oplus \mathbf{3} \oplus \mathbf{3'} \\
\mathbf{9} & = & \mathbf{1} \oplus \mathbf{2} \oplus \mathbf{3} \oplus \mathbf{3'}
\end{array}
\tag{A.1}
$$

We will label the components of $S_4$ representations using $SO(3)$ vector indices as follows. The three-dimensional standard representation of $S_4$ carries an $SO(3)$ vector index $i$, or equivalently, an antisymmetric pair of indices $[jk]$.[11] Similarly, the irreducible representations of $S_4$ can be expressed in terms of the following tensors:

$$
\begin{array}{rclll}
\mathbf{1} & : & S & & \\
\mathbf{1'} & : & T_{(ijk)} & , \quad i \neq j \neq k & \\
\mathbf{2} & : & B_{[ij]k} & , \quad i \neq j \neq k & , \quad B_{[ij]k} + B_{[jk]i} + B_{[ki]j} = 0 \\
& & B_{i(jk)} & , \quad i \neq j \neq k & , \quad B_{i(jk)} + B_{j(ki)} + B_{k(ij)} = 0 \\
\mathbf{3} & : & V_i & & \\
\mathbf{3'} & : & E_{ij} & , \quad i \neq j & , \quad E_{ij} = E_{ji}
\end{array}
\tag{A.2}
$$

In the above we have two different expressions, $B_{[ij]k}$ and $B_{i(jk)}$, for the irreducible representation $\mathbf{2}$ of $S_4$. In the first expression, $B_{[ij]k}$ is the component of $\mathbf{2}$ in the tensor product $\mathbf{3} \otimes \mathbf{3} = \mathbf{1} \oplus \mathbf{2} \oplus \mathbf{3} \oplus \mathbf{3'}$. In the second expression, $B_{i(jk)}$ is the component of $\mathbf{2}$ in the tensor

---

[11]We will adopt the convention that indices in the square brackets are antisymmetrized, whereas indices in the parentheses are symmetrized. For example, $A_{[ij]} = -A_{[ji]}$ and $A_{(ij)} = A_{(ji)}$.

product $\mathbf{3} \otimes \mathbf{3}' = \mathbf{1}' \oplus \mathbf{2} \oplus \mathbf{3} \oplus \mathbf{3}'$. The two bases of tensors are related as[12]

$$
\begin{aligned}
B_{i(jk)} &= B_{[ij]k} + B_{[ik]j}\,, \\
B_{[ij]k} &= \frac{1}{3}\left(B_{i(jk)} - B_{j(ik)}\right).
\end{aligned}
\tag{A.3}
$$

# B  Exotic Global Symmetries

In this appendix we review the tensor and dipole global symmetries of [2]. We will first discuss the $U(1)$ versions of these symmetries and their currents, and then generalize to $\mathbb{Z}_N$. The space is assumed to be a 3-torus with lengths $\ell^x, \ell^y, \ell^z$.

## B.1  Tensor Global Symmetry

The $U(1)$ versions of the following two tensor global symmetries are realized in the $\hat{\phi}$ and the $A$ theories in [2].

$$(\mathbf{2}, \mathbf{3}')\ \textit{Tensor Symmetry}$$

Consider the $U(1)$ tensor symmetry with currents $(J_0^{[ij]k}, J^{ij})$ in the $(\mathbf{2}, \mathbf{3}')$ representations. The current conservation equation is

$$
\partial_0 J_0^{[ij]k} = \partial^i J^{jk} - \partial^j J^{ik}\,.
\tag{B.1}
$$

The $U(1)$ symmetry operator is the exponentiation of the conserved charge $Q$:

$$
\begin{aligned}
\mathcal{U}^{ij}(x_1^k, x_2^k) &= \exp\left[i\beta \int_{x_1^k}^{x_2^k} dx^k Q^{[ij]}\right] \\
&= \exp\left[i\beta \int_{x_1^k}^{x_2^k} dx^k \oint dx^i \oint dx^j\, J_0^{[ij]k}\right]\,, \quad (\text{no sum in } i, j, k)
\end{aligned}
\tag{B.2}
$$

which is a "slab" of width $x_2^k - x_1^k$ in the $k$ direction and extends along the $i, j$ directions. Since $J_0^{[xy]z} + J_0^{[yz]x} + J_0^{[zx]y} = 0$,

$$
\mathcal{U}^{xy}(0, \ell^z)\,\mathcal{U}^{yz}(0, \ell^x)\,\mathcal{U}^{zx}(0, \ell^y) = 1\,.
\tag{B.3}
$$

---

[12]There is a third expression for the $\mathbf{2}$: $B_{ii}$ with $B_{xx} + B_{yy} + B_{zz} = 0$. Repeated indices are not summed over here. This expression is most natural if we embed the $\mathbf{2}$ of $S_4$ into the $\mathbf{5}$ of $SO(3)$ (i.e. symmetric, traceless rank-two tensor). It is related to the first expression $B_{[ij]k}$ as $B_{[ij]k} = \epsilon_{ijk} B_{kk}$.

On a lattice, we have $L^x + L^y + L^z - 1$ such symmetry operators.

If the symmetry group is $\mathbb{Z}_N$ as opposed to $U(1)$, then there are no currents but only the symmetry operators $\mathcal{U}^{ij}(x_1^k, x_2^k)$ with $\beta = \frac{2\pi n}{N}$ and $n = 1, \cdots, N$.

## $(\mathbf{3'}, \mathbf{2})$ *Tensor Symmetry*

Next, consider a different $U(1)$ tensor global symmetry with currents $(J_0^{ij}, J^{[ij]k})$ in the $(\mathbf{3'}, \mathbf{2})$ representations. The currents obey the conservation equation

$$\partial_0 J_0^{ij} = \partial_k (J^{[ki]j} + J^{[kj]i}), \tag{B.4}$$

and the differential constraint

$$\partial_i \partial_j J_0^{ij} = 0. \tag{B.5}$$

The $U(1)$ symmetry operator is a line extended in the $k$ direction at a fixed point $(x^i, x^j)$ on the $ij$-plane:

$$\mathcal{U}^k(x^i, x^j) = \exp\left[ i\beta\, Q^k(x^i, x^j) \right] = \exp\left[ i\beta \oint dx^k\, J_0^{ij} \right]. \tag{B.6}$$

The differential condition (B.5) implies that the position dependence of $\mathcal{U}^k(x^i, x^j)$ factorizes

$$\mathcal{U}^k(x^i, x^j) = \mathcal{U}_i^k(x^i)\mathcal{U}_j^k(x^j). \tag{B.7}$$

and only the product of their zero modes is physical. On a lattice, we have $2L^x + 2L^y + 2L^z - 3$ such symmetry operators.

If the symmetry group is $\mathbb{Z}_N$ as opposed to $U(1)$, then there are no currents but only the symmetry operators $\mathcal{U}^k(x^i, x^j)$ with $\beta = \frac{2\pi n}{N}$ and $n = 1, \cdots, N$.

## $(\mathbf{3'}, \mathbf{2})$ *Unconstrained Tensor Symmetry*

We can also consider a variant of the $(\mathbf{3'}, \mathbf{2})$ tensor symmetry where the differential constraint (B.5) is relaxed. Consequently, the symmetry operator does not obey (B.7). We will call this variant the $(\mathbf{3'}, \mathbf{2})$ unconstrained tensor symmetry. Such a $\mathbb{Z}_N$ unconstrained tensor symmetry is present in the X-cube model (see Section 5.2). The relation between the unconstrained tensor and the tensor symmetries is analogous to that between the non-relativistic [6] and the relativistic one-form symmetries [29].

## B.2 Dipole Global Symmetry

The $U(1)$ versions of the following two dipole global symmetries are realized in the $\phi$ and the $\hat{A}$ theories in [2].

### $(\mathbf{1}, \mathbf{3}')$ Dipole Symmetry

Consider the $U(1)$ dipole symmetry generated by currents $(J_0, J^{ij})$ in the $(\mathbf{1}, \mathbf{3}')$ of $S_4$. They obey

$$
\begin{aligned}
\partial_0 J_0 &= \frac{1}{2}\partial_i\partial_j J^{ij} \\
&= \partial_x\partial_y J^{xy} + \partial_z\partial_x J^{zx} + \partial_y\partial_z J^{yz} \, .
\end{aligned}
\tag{B.8}
$$

The $U(1)$ symmetry operator is a slab with finite width $x_2^k - x_1^k$ in the $k$ direction and extend in the $ij$ direction

$$
\mathcal{U}_{ij}(x_1^k, x_2^k) = \exp\left[i\beta \int_{x_1^k}^{x_2^k} dx^k \, Q_{ij}(x^k)\right] = \exp\left[i\beta \int_{x_1^k}^{x_2^k} dx^k \oint dx^i \oint dx^j \, J_0\right] \, .
\tag{B.9}
$$

They obey

$$
\mathcal{U}_{yz}(0, \ell^x) = \mathcal{U}_{zx}(0, \ell^y) = \mathcal{U}_{xy}(0, \ell^z) \, .
\tag{B.10}
$$

On a lattice, we have $L^x + L^y + L^z - 2$ such symmetry operators.

If the symmetry group is $\mathbb{Z}_N$ as opposed to $U(1)$, then there are no currents but only the symmetry operators $\mathcal{U}_{ij}(x_1^k, x_2^k)$ with $\beta = \frac{2\pi n}{N}$ and $n = 1, \cdots, N$.

### $(\mathbf{3}', \mathbf{1})$ Dipole Symmetry

The second $U(1)$ dipole symmetry is generated by currents $(J_0^{ij}, J)$ with $(\mathbf{3}', \mathbf{1})$ of $S_4$. They obey the conservation equation:

$$
\partial_0 J_0^{ij} = \partial^i\partial^j J \, ,
\tag{B.11}
$$

and a differential condition

$$
\partial^i J_0^{jk} = \partial^j J_0^{ik} \, .
\tag{B.12}
$$

The $U(1)$ symmetry operator is a strip operator:

$$
\mathcal{U}(z_1, z_2, \mathcal{C}^{xy}) = \exp\left[i\beta \int_{z_1}^{z_2} dz \, Q(\mathcal{C}^{xy}, z)\right] = \exp\left[i\beta \int_{z_1}^{z_2} dz \oint_{\mathcal{C}^{xy}} (dx\, J_0^{zx} + dy\, J_0^{yz})\right] \, .
\tag{B.13}
$$

Here the strip is the direct product of the segment $[z_1, z_2]$ and a closed curve $\mathcal{C}^{xy}$ on the $xy$-plane. The differential condition (B.12) implies that the symmetry operator is independent of small deformation of the curve $\mathcal{C}^{xy}$. In other words, the dependence on $\mathcal{C}^{xy}$ is topological.

Similarly, we have symmetry operators along the other directions. They obey

$$
\begin{aligned}
\mathcal{U}(0, \ell^z, \mathcal{C}_x^{xy}) &= \mathcal{U}(0, \ell^x, \mathcal{C}_z^{yz}) \,, \\
\mathcal{U}(0, \ell^z, \mathcal{C}_y^{xy}) &= \mathcal{U}(0, \ell^y, \mathcal{C}_z^{xz}) \,, \\
\mathcal{U}(0, \ell^x, \mathcal{C}_y^{yz}) &= \mathcal{U}(0, \ell^y, \mathcal{C}_x^{xz}) \,,
\end{aligned}
\tag{B.14}
$$

where $\mathcal{C}_i^{ij}$ is any closed curve on the $ij$-plane that wraps around the $i$ direction once but not the $j$ direction. On a lattice, we have $2L^x + 2L^y + 2L^z - 3$ such symmetry operators.

If the symmetry group is $\mathbb{Z}_N$ as opposed to $U(1)$, then there are no currents but only the symmetry operators $\mathcal{U}(z_1, z_2, \mathcal{C}^{xy})$ with $\beta = \frac{2\pi n}{N}$ and $n = 1, \cdots, N$.

$$(\mathbf{3}', \mathbf{1}) \;\; \textit{Unconstrained Dipole Symmetry}$$

We can also consider a variant of the $(\mathbf{3}', \mathbf{1})$ dipole symmetry where the differential constraint (B.12) is relaxed. Consequently, the symmetry operator depends on the detailed shape of the curve $\mathcal{C}^{ij}$. We will call this variant the $(\mathbf{3}', \mathbf{1})$ unconstrained dipole symmetry. Such a $\mathbb{Z}_N(\mathbf{3}', \mathbf{1})$ unconstrained dipole symmetry is present in the X-cube model (see Section 5.2). Again, the relation between the unconstrained dipole and the dipole symmetries is analogous to that between the non-relativistic [6] and the relativistic one-form symmetries [29].

# C $\;\;$ $\mathbb{Z}_N$ Gauge Theory and Toric Code

This appendix reviews various presentations of ordinary $\mathbb{Z}_N$ gauge theories. The purpose of this review is to demonstrate, in a well-known setting, the various approaches that we use in the body of the paper when we study more sophisticated $\mathbb{Z}_N$ gauge theories.

## C.1 $\;\;$ The Lattice Model

### $\mathbb{Z}_N$ Lattice Gauge Theory

We start with a Euclidean $(D + 1)$-dimensional cubic lattice, whose sites are labeled by integers $(\hat{\tau}, \hat{x}, \hat{y}, \hat{z} \cdots)$. The degrees of freedom $U_\mu$ are $\mathbb{Z}_N$ group elements on the links. The

gauge transformation parameters are $\mathbb{Z}_N$ elements $\eta(\hat{\tau}, \hat{x}, \hat{y}, \hat{z} \cdots)$ on the sites and they act on $U_\mu$ as

$$U_x(\hat{\tau}, \hat{x}, \hat{y}, \hat{z} \cdots) \to U_x(\hat{\tau}, \hat{x}, \hat{y}, \hat{z} \cdots) \eta(\hat{\tau}, \hat{x}, \hat{y}, \hat{z} \cdots) \eta(\hat{\tau}, \hat{x}+1, \hat{y}, \hat{z} \cdots)^{-1} \qquad \text{(C.1)}$$

and similarly for $U_y$, etc. The simplest gauge invariant interaction involves an oriented product of $U_\mu$ around a plaquettes

$$L_{xy}(\hat{\tau}, \hat{x}, \hat{y}, \hat{z} \cdots) = U_x(\hat{\tau}, \hat{x}, \hat{y}, \hat{z} \cdots) U_y(\hat{\tau}, \hat{x}+1, \hat{y}, \hat{z} \cdots) U_x(\hat{\tau}, \hat{x}, \hat{y}+1, \hat{z} \cdots)^{-1} U_y(\hat{\tau}, \hat{x}, \hat{y}, \hat{z} \cdots)^{-1}$$
$$\text{(C.2)}$$

More complicated interactions are also possible.

The $\mathbb{Z}_N$ lattice gauge theory includes magnetic excitations, but it has no electric excitations. Correspondingly, it does not have a magnetic global symmetry, but it does have an electric one-form global symmetry [29]. The one-form symmetry multiplies all the link variables by arbitrary $\mathbb{Z}_N$ phases $h_\mu(\hat{\tau}, \hat{x}, \hat{y}, \hat{z} \cdots)$ such that their oriented product around every plaquette, as in (C.2), is one. Most of these transformations are gauge transformations, but those that are not gauge transformations act as a global symmetry. The objects charged under this $\mathbb{Z}_N$ one-form global symmetry are $\mathbb{Z}_N$ Wilson lines – products of $U_\mu$s along closed curves.

In the Hamiltonian formulation, we use the analog of temporal gauge setting all the links in the time direction to one, i.e. $U_\tau = 1$. We also introduce "momenta" $V_i$ conjugate to $U_i$ on the links. ($\mu$ was a Euclidean spacetime direction and $i$ is a spatial direction.) They are conjugate variables in the sense that $U_i$ and $V_i$ on the same link satisfy a Heisenberg algebra

$$U_i V_i = e^{2\pi i/N} V_i U_i \qquad \text{(C.3)}$$

and elements on different links commute. In addition, we need to impose Gauss law. It is an operator constraint at every site given by

$$G(\hat{x}, \hat{y}, \hat{z} \cdots) = V_x(\hat{x}, \hat{y}, \hat{z} \cdots) V_x(\hat{x}-1, \hat{y}, \hat{z} \cdots)^{-1} V_y(\hat{x}, \hat{y}, \hat{z} \cdots) V_y(\hat{x}, \hat{y}-1, \hat{z} \cdots)^{-1} \cdots = 1 \ ,$$
$$\text{(C.4)}$$

where the product of $V_i$s includes all the link variables connected to the site $(\hat{x}, \hat{y}, \hat{z}, \cdots)$.

## Toric Code

It is common, following [35,36], not to impose Gauss law (C.4) as an operator constraint, but instead, to add a term to the Hamiltonian to raise the energy of states violating it. The

simplest such Hamiltonian is the toric code:

$$H_{\text{toric}} = -\sum_{\text{sites}} G - \sum_{\text{plaquettes}} L_{ij} + c.c.. \tag{C.5}$$

The first term imposes the Gauss law energetically. The low-lying states satisfy $G(\hat{x}, \hat{y}, \hat{z} \cdots) = 1$, but excited states do not satisfy it. Once such a term is added to the Hamiltonian there is no need to preserve the underlying gauge symmetry and more interactions can be added, e.g. $\sum_i U_i$.

The toric code includes both electrically-charged and magnetically-charged dynamical excitations. Therefore, it does not have global electric or magnetic generalized symmetries of the kind studied in [29]. Instead, it has the non-relativistic electric and magnetic one-form global symmetries studied in [6]. If additional terms, e.g. $\sum_i U_i$, are added to the Hamiltonian, even the non-relativistic electric symmetry is violated [6]. Similarly, additional terms like $\sum_i V_i$ break the non-relativistic magnetic symmetry.

These two lattice systems, the ordinary lattice $\mathbb{Z}_N$ gauge theory and the toric code, have the same low-energy limit. In other words, they are dual to each other in the infrared. We will now discuss this continuum $\mathbb{Z}_N$ gauge theory.

## C.2    Continuum Lagrangians

*The Three Dual Descriptions of the Continuum Theory*

There are several presentations of the continuum $\mathbb{Z}_N$ gauge theory. One presentation is in terms of an ordinary $U(1)$ gauge theory with a gauge field $A$ (which is locally a one-form) coupled to a charge-$N$ scalar Higgs field $\phi$. The gauge group is then Higgsed from $U(1)$ to $\mathbb{Z}_N$. The Lagrangian is[13]

$$\mathcal{L}_{A/\phi} = \frac{i}{2\pi} \hat{F}(d\phi - NA). \tag{C.6}$$

Here $\hat{F}$ is an independent $D$-form field. It acts as a Lagrange multiplier setting $d\phi - NA = 0$. This Higgses $U(1)$ to $\mathbb{Z}_N$. This presentation of the $\mathbb{Z}_N$ gauge theory is similar to the one used in Section 2.

Instead of using (C.6), we can follow [30–32,29] and dualize the scalar $\phi$ to a $(D-1)$-form gauge field. The resulting Lagrangian is

$$\mathcal{L}_{A/\hat{A}} = \frac{N}{2\pi} \hat{A} dA, \tag{C.7}$$

---

[13]Since unlike in the body of the paper this system is relativistic, we use form notation.

where $\hat{A}$ is a $(D-1)$-form gauge field.[14] It is related to $\hat{F}$ in (C.6) through $\hat{F} = d\hat{A}$. This presentation of the theory is similar to the one in Section 4.

We can also further dualize $A$ to a $(D-2)$-form gauge field $\hat{\phi}$ to convert (C.7) to

$$\mathcal{L}_{\hat{A}/\hat{\phi}} = \frac{i}{2\pi} F(d\hat{\phi} - N\hat{A}) \,. \tag{C.8}$$

Here $F$ is a two-form field. It is a Lagrangian multiplier enforcing $d\hat{\phi} - N\hat{A}$. In this presentation, the $U(1)$ gauge symmetry of $\hat{A}$ is Higgsed to $\mathbb{Z}_N$.

This presentation of the theory is similar to the one in Section 3.

It should be noted that the presentation (C.8) motivates another lattice construction of the same system. Here the continuum gauge field $\hat{A}$ is replaced by a gauge field on $(D-1)$-dimensional cubes, whose gauge parameters take values on $(D-2)$-dimensional cubes. This is similar to the discussion in Section 3.1.

### Defects and Operators

The continuum $\mathbb{Z}_N$ gauge theory has neither electric nor magnetic excitations. Such excitations, if present, have high energy of the order to the lattice scale. Therefore, they are effectively classical. This means that the low-energy effective theory has both an electric and a magnetic generalized global symmetry [29].

The physics of massive probes charged under the above generalized global symmetries is captured by operators/defects. This is most clear in the presentation (C.7), where the natural observables are

$$W_E = e^{i\oint A} \quad \text{and} \quad W_M = e^{i\oint \hat{A}} \,. \tag{C.9}$$

Here, the integrals are over a 1-dimensional curve and a $(D-1)$-dimensional manifold in spacetime, respectively. These represent electric and the magnetic objects .

If such operators act at the same time and they pierce each other, they do not commute. Therefore, $W_E$ is the operator generating the magnetic symmetry and $W_M$ is the operator generating the electric symmetry. See [29] for more details. Also, since these operators act in the space of ground states, we can interpret these global symmetries as being spontaneously broken.[15] We emphasize that even when the lattice system does not have these global symmetries, these symmetries arise as accidental symmetries acting in the low-energy theory.

---

[14]Often, this Lagrangian is written as $\frac{N}{2\pi} BF = \frac{N}{2\pi} BdA$ and hence the name $BF$-theory. Since we use the letter $B$ for a magnetic field, we write it in terms of $\hat{A}$.

[15]In quantum information theory such operators are referred to as logical operators. We thank M. Hermele for a helpful discussion about it.

Finally, when $D > 1$ the low-energy theory does not have any local, gauge-invariant operators. Therefore, it is robust. (See [1].) Its accidental higher-form global symmetries cannot be ruined and the structure of the ground states cannot be changed by any short-distance perturbation, provided it is small enough. This is similar to the $(3+1)$-dimensional continuum $\mathbb{Z}_N$ tensor gauge theory of this paper.

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
