# Peer review of "Exotic $\mathbb{Z}_N$ Symmetries, Duality, and Fractons in 3+1-Dimensional Quantum Field Theory"

_SciPost Physics_

## Round 2 · Referee Report · Anonymous (Referee 1) · 2020-11-7

Strengths

This paper describes a continuum Lagrangian with some rather unusual properties for a gapped fracton phase of matter. This is a timely subject area and the authors are bringing a high energy physics perspective to this problem originating in condensed matter. Since the work has certainly been of interest to the community, I have no objection to publication after minor changes.

Weaknesses

The entire paper is extremely technical. Even for people in adjacent fields, but who did not carefully read these authors' earlier papers, e.g. Ref. [2], this paper takes a lot of effort to comprehend. It's clear that this paper represents a continuation of earlier papers, and unfortunately I expect one has to read at least one of the two to get very much out of this paper. I give a few suggestions below for where the authors could try to make this a tiny bit more self contained, though ultimately I doubt it is a good use of the authors' time. Likely it is already useful to some specialists in the field, even if it is very difficult for outsiders to follow.

Report

The main purpose of the paper, as far as I can tell, appears to be a field theoretic understanding of the ground state degeneracy of a Z_N variant of the X-cubed model. Unsurprisingly, the low energy effective theory is not dynamical and its equations of motion reduce to a set of simple equations. The authors presented 3 dual versions of this field theory. While at first this seemed a little surprising, I think after reading Section 5.1 it felt clear that these theories were dual.

I understand the perspective of the authors which is to try and explain how fracton matter can be explained in a field theoretic language. While I can't say for sure, it does sound very nice to have a crisp understanding of the ground state degeneracy of the X-cubed models, along with presumably some other microscopic models with similar symmetries. However, it did seem a little awkward to me to use a field theory to describe something that is infinite without a lattice discretization. Typically one might like to understand some other correlation functions that are insensitive to lattice details. I am not sure the extent to which that is possible in this Z_N theory (in the U(1) case I am a little more optimistic).

While I don't know how promising this direction will end up being, the authors are clearly quite knowledgeable in QFT and their insight into this problem seems worth publishing. My comments are essentially all about readability, though eventually it felt pointless to keep track of the number of things that wee introduced with almost zero motivation/explanation. If the authors wanted to make the paper more self-contained and approachable I think it would be clear how to do it.

Requested changes

1) It looks like there is a typo in (2.3) on indices.

2) In the introduction the authors talk about a (1,3') etc. tensor symmetry. This is never explained, after seeing some equations it was clear what that meant but I didn't feel that was obvious or standard notation in the community. It should be introduced clearly.

3) I found the "momentum symmetry" statement to be a little strange. Might be a culture clash between different fields. I think I get what the authors mean from the table...

4) There were other papers in the literature that used field theoretic constructions with mixed rank tensors to understand aspects of fracton physics. I think in some cases people tried to understand various correlation functions etc.; certainly the mobility of charges was tied to the different tensor structures in the past. I think it might be helpful if the authors more clearly delineated what is new in this work. Is it primarily the duality between the 3 quadratic Lagrangians in Secs. 2-4? Is there something new in this work for a condensed matter physicist? I am not objecting to citations or asking for any other citations in any way, but I just think that some clarity might be called for here.

---

## Round 2 · Referee Report · Anonymous (Referee 2) · 2020-12-13

Strengths

1- The paper gives a clear exposition of a field theory approach to abelian fracton models with cubic symmetry.

2- This paper in the series is probably best understood as an illustration of the general formalism being developed by the authors showing that it can reproduce physics familiar from solvable lattice models.

Weaknesses

1- Though I strongly support the program the authors are pursuing, I was struggling to overcome the impression that this particular paper is largely a transcription of known results about the X-cube model into a different notation. Of course it is a necessary check of the formalism that it can reproduce the known physics of the X-cube model. I would have liked to see also some conclusion about physics (rather than about the development of the QFT formalism) that I didn't know already. (If it was there and I missed it, please emphasize it more.)

2- A small issue of presentation: in the discussions in sections 2.1 and 3.1 of lattice models, the authors overemphasize the distinction between the solvable fixed-point hamiltonian for the X-cube model and the family of lattice hamiltonians describing perturbations within the phase (parameterized by $g_e$ and $\hat g_e$). As the authors understand well, the reason that studies of the X-cube model focus on the solvable limit is so that they can solve it. On the other hand, we also know that there is an open set around that limit with the same universal properties. So I think it is best not to be too dogmatic about distinguishing between "the X-cube model" and other models in the phase it represents.

3- In the caption to figure 1, it is not clear what one means by saying that a field theory "is the higgs field of" another field theory.

4- As with other papers in the series, I think the distinctions between the analysis here and existing work on field theory descriptions of fractons are somewhat overstated. The authors often cite this other work by saying "aspects of... were studied" in a way which is not always fair to that prior work. It is true that the present work is more systematic and has a clearer starting viewpoint, in a way that suggests many directions for progress which were previously obscure. Why not try a little harder to keep everyone happy?

Report

First, I apologize for the delay.

This paper is part of a series of several papers by the authors attempting to generalize continuum quantum field theory to account for fracton topological phases.
The paper should be published.

Requested changes

Please see the "weaknesses" section.

---

## Editorial Decision

resubmitted